# Rapid movement and transcriptional re-localization of human cohesin on DNA

Iain F Davidson[1], Daniela Goetz[1,†], Maciej P Zaczek[1,†], Maxim I Molodtsov[1,2,†], Pim J Huis in 't Veld[1,‡], Florian Weissmann[1], Gabriele Litos[1], David A Cisneros[1,§], Maria Ocampo-Hafalla[3], Rene Ladurner[1,¶], Frank Uhlmann[3], Alipasha Vaziri[1,2,4,††] & Jan-Michael Peters[1,*]

## Abstract

The spatial organization, correct expression, repair, and segregation of eukaryotic genomes depend on cohesin, ring-shaped protein complexes that are thought to function by entrapping DNA. It has been proposed that cohesin is recruited to specific genomic locations from distal loading sites by an unknown mechanism, which depends on transcription, and it has been speculated that cohesin movements along DNA could create three-dimensional genomic organization by loop extrusion. However, whether cohesin can translocate along DNA is unknown. Here, we used single-molecule imaging to show that cohesin can diffuse rapidly on DNA in a manner consistent with topological entrapment and can pass over some DNA-bound proteins and nucleosomes but is constrained in its movement by transcription and DNA-bound CCCTC-binding factor (CTCF). These results indicate that cohesin can be positioned in the genome by moving along DNA, that transcription can provide directionality to these movements, that CTCF functions as a boundary element for moving cohesin, and they are consistent with the hypothesis that cohesin spatially organizes the genome via loop extrusion.

**Keywords** cell cycle; cohesin; genome organization; single-molecule TIRF microscopy; transcription
**Subject Categories** Cell Cycle; Chromatin, Epigenetics, Genomics & Functional Genomics; Transcription
**The EMBO Journal (2016) 35: 2671–2685**

## Introduction

Cohesin complexes mediate sister chromatid cohesion, which is essential for proper chromosome segregation in dividing cells, but also have important roles in DNA damage repair, recombination, higher-order chromatin structure, and gene regulation in both proliferating and quiescent cells (reviewed in Merkenschlager & Nora, 2016). The cohesin core complex is composed of four subunits. Three of these, Smc1, Smc3, and Scc1 (also called Rad21 or Mcd1), assemble into tripartite rings with an inner diameter of ~35 nm (Anderson *et al*, 2002; Haering *et al*, 2002; Gruber *et al*, 2003; Ivanov & Nasmyth, 2005; Haering *et al*, 2008). The fourth subunit, in mammalian somatic cells either Stag1 or Stag2, is bound to Scc1. Related "structural maintenance of chromosomes" (SMC) complexes control the organization of mitotic chromosomes (condensin complexes) and bacterial genomes (Hirano, 2016).

To perform its functions, cohesin has to associate with DNA. *In vivo*, this interaction depends on ATP hydrolysis by Smc1 and Smc3, and on the Nipbl/Mau2 (also known as Scc2/Scc4) cohesin loading complex (Ciosk *et al*, 2000; Arumugam *et al*, 2003; Weitzer *et al*, 2003; Gillespie & Hirano, 2004; Takahashi *et al*, 2004; Watrin *et al*, 2006; Hu *et al*, 2011; Ladurner *et al*, 2014) and can be reversed either by the cohesin-associated protein Wapl or the protease separase. Both of these are thought to open the cohesin ring (Huis in 't Veld *et al*, 2014 and references therein). Experimentally, cohesin–DNA interactions can also be reversed by cleavage of cohesin, or alternatively by cleavage of DNA (Gruber *et al*, 2003; Ivanov & Nasmyth, 2005). This phenomenon and yeast mini-chromosome experiments (Haering *et al*, 2008) indicate that cohesin entraps DNA inside its ring. It has been proposed that cohesin uses this ability to mediate both sister chromatid cohesion and chromatin loop formation. According to this hypothesis, cohesin would generate cohesion by entrapping two sister DNA molecules (Gruber *et al*, 2003; Ivanov & Nasmyth, 2005; Haering *et al*, 2008), but would form chromatin loops by encircling two regions of the same DNA molecule (Hadjur *et al*, 2009; Nativio *et al*, 2009). The latter interactions are thought to contribute to gene regulation by controlling the proximity between enhancer and promoter sequences, and to enable

1 Research Institute of Molecular Pathology (IMP), Vienna, Austria
2 Max F. Perutz Laboratories, University of Vienna, Vienna, Austria
3 The Francis Crick Institute, London, UK
4 The Rockefeller University, New York, NY, USA
  *Corresponding author. Tel: +43 1797303000; E-mail: peters@imp.ac.at
  †These authors contributed equally to this work
  ‡Present address: Department of Mechanistic Cell Biology, Max Planck Institute of Molecular Physiology, Dortmund, Germany
  §Present address: The Laboratory for Molecular Infection Medicine Sweden (MIMS) and Department of Molecular Biology, Umeå University, Umeå, Sweden
  ¶Present address: Department of Biochemistry, Stanford University, Stanford, CA, USA
  ††The affiliations have been corrected on 15 November 2016 after first online publication
  [The copyright line of this article was changed on 6 February 2017 after first online publication.]

recombination events (Rollins *et al*, 1999; Kagey *et al*, 2010; Guo *et al*, 2011; Seitan *et al*, 2011; Medvedovic *et al*, 2013; Seitan *et al*, 2013). Cohesin performs these functions together with CCCTC-binding factor (CTCF), a zinc-finger DNA binding protein that recognizes specific sequences in the genome and with which cohesin co-localizes at most of its binding sites in mammalian genomes (Parelho *et al*, 2008; Wendt *et al*, 2008).

Although cohesin is enriched at particular sites in the genome and is thought to mediate interactions between specific pairs of these, several observations indicate that the distribution of cohesin in the genome is highly dynamic. In yeast cells, cohesin is recruited to DNA by the cohesin loading complex at sites that are distinct from most of its final binding sites and can be relocated to 3′ ends of active genes by transcription (Glynn *et al*, 2004; Lengronne *et al*, 2004; Schmidt *et al*, 2009; Hu *et al*, 2011; Ocampo-Hafalla *et al*, 2016). Also in mammalian cells, the cohesin loading complex has been detected at genomic sites that are distinct from cohesin sites (Kagey *et al*, 2010; Zuin *et al*, 2014b), consistent with the possibility that cohesin can also be relocated within mammalian genomes. A distinct type of cohesin translocation has been proposed to explain how cohesin and CTCF are able to bring specific sequences into close proximity to mediate the formation of chromatin loops. According to this hypothetical model, distant but defined sequences on a chromosome would be brought into proximity by cohesin which would extrude a chromatin loop until it either encounters boundary elements, such as CTCF bound to its cognate binding sequences, or until it is released from DNA (Nichols & Corces, 2015; Sanborn *et al*, 2015; Fudenberg *et al*, 2016). However, it remained unknown whether cohesin is actually able to move along DNA, how transcription can relocate cohesin and how CTCF and other DNA-bound proteins might influence this process.

To address these questions, we visualized cohesin–DNA interactions at the single-molecule level in real time using total internal reflection fluorescence (TIRF) microscopy. This revealed that human cohesin translocates rapidly on DNA, as was also reported during preparation of our manuscript for fission yeast cohesin and the SMC complex from *Bacillus subtilis* (Kim & Loparo, 2016; Stigler *et al*, 2016). Our experiments show further that recombinant human cohesin is released from DNA following DNA or cohesin ring cleavage, but not by high-salt treatment, indicating that cohesin topologically entrapped DNA in our reconstituted system. Under these conditions, cohesin is able to pass over some DNA-bound proteins and nucleosomes but is constrained in its movement by T7 RNA polymerase and CTCF. These results are consistent with the hypotheses that cohesin is positioned in the genome by rapidly moving along DNA via passive diffusion, that transcription can provide directionality to these movements, and that CTCF functions as a boundary element to translocating cohesin.

## Results

### Recombinant human tetrameric cohesin complexes bind to DNA, translocate rapidly in high-salt buffer, and are released following DNA or cohesin cleavage

We first reconstituted the binding of recombinant human cohesin composed of Smc1, Smc3, Scc1, and Stag1 (Fig 1A) to DNA, using a

bulk assay developed by Murayama and Uhlmann (Murayama & Uhlmann, 2014). These authors observed that loading of fission yeast cohesin onto circular DNA is stimulated by the cohesin loading complex and ATP and that the resulting cohesin–DNA interactions are sensitive to Scc1 and DNA cleavage, consistent with topological entrapment. In our experiments, human cohesin could bind a small amount of circular 3.3 kb DNA (around 1–10%, depending on elution method) in the absence of the cohesin loading complex and in a manner that was not enhanced by ATP (Figs EV1A and EV2A–D). This binding was greatly reduced if DNA had been linearized (Fig 1B) or if a form of cohesin was used in which a recognition site for tobacco etch virus (TEV) protease engineered into Scc1 had been cleaved (Fig 1C and D). This treatment opens the cohesin ring (Huis in 't Veld *et al*, 2014) and mimics cohesin cleavage by separase, which initiates sister chromatid separation. Like fission yeast cohesin, a small amount of human cohesin can therefore associate with DNA spontaneously in the absence of the cohesin loading complex in a manner that depends on circularity of cohesin and DNA.

To visualize cohesin, we fused wild-type and TEV protease-cleavable Scc1 to green fluorescent protein (GFP) or HaloTag, which we labeled with tetramethylrhodamine (TMR; Fig EV1B). Cleavage of Scc1$^{Halo-TEV}$ or Scc1$^{GFP-TEV}$ by TEV protease did not displace it from Smc1/Smc3, indicating that the non-cleaved cohesin complexes were ring-shaped (Figs 1C and EV3A; Huis in 't Veld *et al*, 2014). Cohesin containing Scc1$^{GFP}$ or Scc1$^{Halo}$ associated with chromatin in *Xenopus* egg extract and was released following TEV protease-mediated Scc1$^{TEV}$ cleavage (Fig EV1C; Huis in 't Veld *et al*, 2014). HeLa cells in which all *Scc1* alleles were modified by CRISPR-Cas9 to express Scc1$^{GFP}$ were viable and proliferated similarly to wild-type cells (Appendix Fig S1). Cohesin containing fluorescently tagged Scc1 is therefore able to perform its essential cellular functions. We flowed these complexes into microscopy chambers in which linear biotinylated λ-phage DNA (48,502 bp; 16 μm) had been tethered at one or both ends (median tether length 10.6 μm, Appendix Fig S2) to an avidin-modified glass surface (Yardimci *et al*, 2010) and imaged cohesin–DNA interactions using a Zeiss TIRF 3 Axio Observer microscope. In low-salt buffer and in the absence of exogenously added ATP, cohesin bound all DNA and compacted singly but not doubly tethered molecules (Fig EV1D and G). This compaction activity was reminiscent of that reported for yeast Smc1–Smc3 heterodimers (Sun *et al*, 2013), the *Xenopus* condensin I complex (Strick *et al*, 2004), and *Bacillus subtilis* SMC (Kim & Loparo, 2016).

To test whether the observed cohesin–DNA interactions were similar to the ones in cells, we first exposed cohesin bound to λ-DNA to high-salt buffer (750 mM NaCl), which extracts most proteins except cohesin (Ciosk *et al*, 2000; Murayama & Uhlmann, 2014). This removed most cohesin and what remained was pushed by buffer flow to the ends of the doubly tethered DNA, but moved rapidly along DNA upon cessation of flow (Fig EV1E, F and H). When we flowed the restriction enzyme XhoI, which cuts λ-DNA at 33,498 bp, into the microscopy chamber, high-salt-resistant cohesin was rapidly released from 89% of DNA molecules ($n = 104$) (Fig 1E and Appendix Fig S3A). Likewise, cohesin was released when DNA broke spontaneously (Fig EV1F, last frame). Unlike intact cohesin complexes, tetrameric complexes that had been cleaved by TEV protease during purification failed to bind to DNA (Fig EV3A–C). Scc1 cleavage also released cohesin after DNA binding as TEV

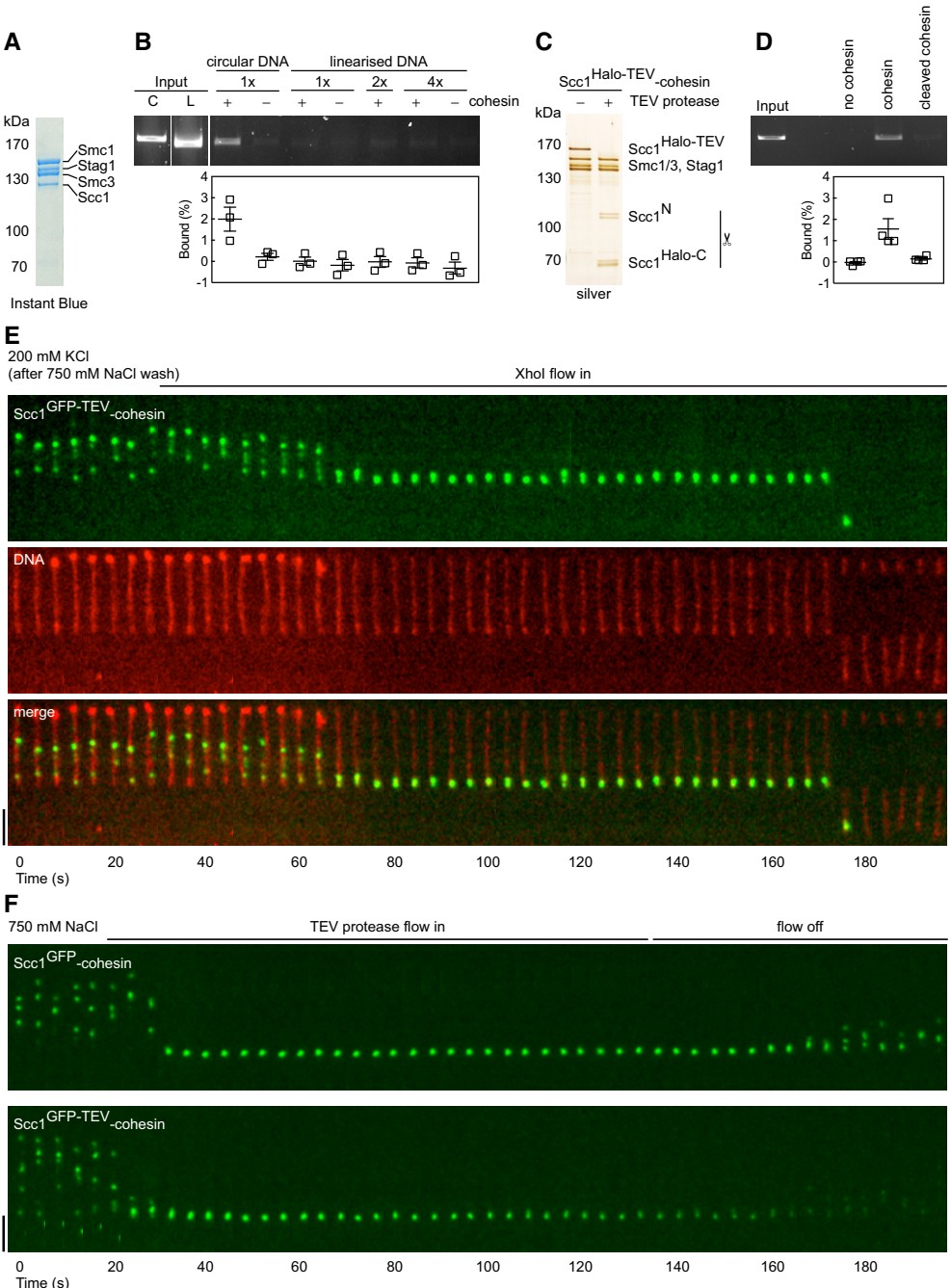

**Figure 1.  Recombinant human tetrameric cohesin complexes bind to DNA, translocate rapidly in high-salt buffer, and are released following DNA or cohesin cleavage.**

A   Instant Blue stained SDS–polyacrylamide gel of Scc1[wt]-cohesin tetramers used in (B).

B   Cohesin loading assay. Scc1[wt]-cohesin tetramer was incubated with nicked circular (C) or 1×, 2×, or 4× concentration of linearized (L) plasmid DNA, immunoprecipitated with anti-Scc1 antibodies, washed with high-salt buffer, and then eluted with Scc1 peptide. Recovered DNA was separated by agarose gel electrophoresis and stained with GelRed DNA stain. Input DNA = 7%. Mean ± SEM is shown.

C   Silver stained SDS–polyacrylamide gel of non-cleaved and cleaved Scc1[Halo-TEV]-cohesin tetramers used in (D).

D   Non-cleaved and cleaved Scc1[Halo-TEV]-cohesin tetramer were incubated with nicked circular plasmid DNA and processed as in (B). Input DNA = 4%. Mean ± SEM is shown.

E   Kymograph of Scc1[GFP-TEV]-cohesin bound to doubly tethered λ-DNA in cohesin binding buffer and washed with 750 mM NaCl buffer + Sytox Orange. XhoI flow in induced DNA cleavage and cohesin release.

F   Kymographs of Scc1[GFP-TEV]-cohesin and Scc1[GFP]-cohesin bound to doubly tethered λ-DNA and washed with 750 mM NaCl buffer. TEV protease flow in released Scc1[GFP-TEV] but not Scc1[GFP] from DNA.

Data information: Flow in from top and scale bar = 5 μm in all kymographs.
Source data are available online for this figure.

protease gradually released high-salt-resistant Scc1[GFP-TEV]-cohesin, but not Scc1[GFP]-cohesin, from DNA, indicating that cohesin opened by Scc1 cleavage cannot persist on DNA (Fig 1F and Appendix Fig S3B). Consistently, Smc1/Smc3[Halo-TMR] dimers also failed to bind to DNA (Fig EV3D–F). In the absence of the cohesin loading complex and exogenously added ATP, some cohesin therefore associates with DNA similarly to how cohesin interacts with DNA in cells, in that these interactions are high-salt resistant and dependent on DNA and cohesin integrity. In subsequent experiments, we only analyzed salt-resistant cohesin on doubly tethered DNA. Although our results do not prove that high-salt-resistant cohesin entraps DNA, they can clearly be explained by and are consistent with this hypothesis.

**Single cohesin complexes bind to DNA and translocate rapidly in high-salt buffer**

Cohesin structures that moved on DNA displayed varying fluorescent intensities, indicating that most contained several molecules (Fig EV1F). Within these structures, most cohesin complexes must interact with DNA similarly, possibly by entrapping DNA, because Scc1 cleavage released cohesin gradually (Fig 1F), and not in one step, as one might have expected if only one cohesin ring had encircled DNA and the others had only been associated with this cohesin molecule. Whether cohesin oligomerization occurs in cells is unknown. We therefore analyzed if single molecules could be detected on a custom-built TIRF microscope with higher optical sensitivity and temporal resolution. Here, some cohesin structures bleached in one step (Fig 2A–C), indicating they carried a single fluorophore and therefore represented single molecules. These complexes moved rapidly on DNA, with a diffusion coefficient of

$1.72 \pm 0.1$ µm$^2$/s (Fig 2D). This is up to four orders of magnitude higher than the diffusion coefficients of many other DNA binding proteins (Gorman & Greene, 2008) and is similar to that of human PCNA and fission yeast cohesin sliding on naked DNA (Kochaniak et al, 2009; Stigler et al, 2016). Movements were not only seen in 750 mM NaCl but also in more physiological salt concentrations (75 mM NaCl and 75 mM KCl, see below), ruling out high-salt artifacts. The unusual rate at which this movement occurs further supports the hypothesis that cohesin entraps DNA. The diffusion coefficients of wild type and ATP binding-mutant forms of cohesin were similar in the presence and absence of ATP, suggesting that ATP is not required for cohesin translocation (Figs 2D and EV2E).

**Cohesin bypasses DNA-bound TetR[Halo-TMR], [Halo-TMR]EcoRI[E111Q], and TMR-labeled nucleosomes but not [QDot]EcoRI[E111Q]**

In cells, most DNA is assembled into nucleosomal 10-nm chromatin fibers. We therefore tested if cohesin can also move on DNA associated with other proteins. First, we bound Halo-tagged bacterial Tet repressor (TetR) to 26,192-bp DNA containing seven Tet operator sequences (pPlat-TetO) (Appendix Figs S4A and B, and S5A and B; the diameter of dimeric DNA-bound TetR excluding HaloTags is ~7 nm), flowed in Scc1[GFP-TEV]-cohesin, and washed with "medium-salt buffer" (75 mM NaCl and 75 mM KCl) to increase cohesin diffusion because high salt would have disrupted TetR–DNA interactions. Under these conditions, 27 out of 37 diffusing Scc1[GFP-TEV]-cohesin structures passed TetR[Halo-TMR] in 160 s (Fig 3A). Similar observations were made when we analyzed cohesin on λ-DNA bound by catalytically inactive, Halo-tagged EcoRI[E111Q] (Appendix Figs S4C and D, and S5A and C; the diameter

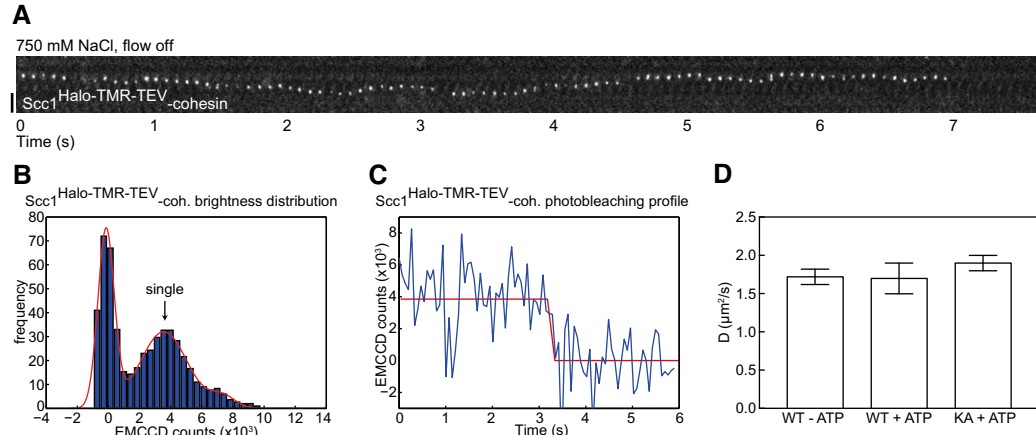

**Figure 2.  Single cohesin complexes bind to DNA and translocate rapidly in high-salt buffer.**

A Kymograph of a single DNA-bound Scc1[Halo-TMR-TEV]-cohesin complex after 750 mM NaCl buffer wash.

B Distribution of background subtracted EMCCD counts of Scc1[Halo-TMR-TEV]-cohesin complexes immobilized on the coverslip. Peaks at ~4 × 10$^3$ and ~8 × 10$^3$ EMCCD counts correspond to single fluorophores and a small fraction of double fluorophores, respectively. $n = 290$ regions with fluorescent molecules, 228 regions with background.

C Photobleaching kinetics of a single DNA-bound Scc1[Halo-TMR-TEV]-cohesin complex.

D Diffusion coefficients of Smc1/3 wild-type or K38A ATP binding-deficient "KA" forms of Scc1[Halo-TMR-TEV]-cohesin in the presence or absence of ATP. Diffusion coefficients were calculated based on the linear fit of the average mean square displacement of ≥ 13 freely diffusing molecules per condition over a 1-s time period. Mean ± SEM is shown.

Data information: Scale bar = 5 µm.
Source data are available online for this figure.

of dimeric DNA-bound EcoRI$^{E111Q}$ excluding HaloTags is ~5 nm); 89 out of 98 diffusing Scc1$^{GFP-TEV}$–cohesin structures readily passed $^{Halo-TMR}$EcoRI$^{E111Q}$ bound to the five EcoRI sites on λ-DNA in 160 s in medium-salt buffer (Fig 3B; for high temporal resolution imaging, see Appendix Fig S6A). Under similar conditions (100 mM NaCl), EcoRI binds to DNA with a half-life of 22 h and blocks the passage of other DNA-binding proteins (for references, see Finkelstein *et al*, 2010), implying that cohesin could not simply pass because EcoRI transiently dissociated from DNA. We next prepared TMR-labeled recombinant histone octamers (Appendix Figs S4E and S5E, diameter ~11 nm) and deposited them onto pPlat at random or at a 601 strong positioning sequence (Appendix Fig S4F); 25 out of 40 cohesin structures passed nucleosomes on pPlat, and 33 out of 56 passed nucleosomes on pPlat-601 (Fig 3C). In contrast, cohesin was unable to pass EcoRI$^{E111Q}$ that was immunocoupled to quantum dots (approximate diameter ~21 nm; Appendix Fig S5F). None of 30 Scc1$^{GFP-TEV}$–cohesin structures passed $^{QDot}$EcoRI$^{E111Q}$ in 160 s (Fig 3D; for high temporal resolution imaging, see Appendix Fig S6B), indicating that cohesin can pass DNA-bound proteins similar in size to nucleosomes (~11 nm), but not over larger structures (≥ 21 nm). Cohesin might therefore also be able to move along DNA in cells, possibly without nucleosomes having to be disassembled or cohesin having to be released and reloaded. Consistent with the latter interpretation, we observed that cohesin complexes did not pass over each other, as cohesin structures of different fluorescent intensities never switched positions when moving on DNA (Fig EV1F).

## Transcription and CTCF constrain cohesin translocation

To test whether transcription affects cohesin movement on DNA, we used T7 bacteriophage RNA polymerase (T7 RNAP), a highly processive single-subunit enzyme that is easier to manipulate than eukaryotic RNA polymerases. We first analyzed the effect of T7 RNAP on the *in vivo* genomic distribution of cohesin in budding yeast (Fig EV4; Ocampo-Hafalla *et al*, 2016). We replaced the endogenous promoter of the *GAL2* gene with a T7 promoter and determined the localization of cohesin in the vicinity of this locus using chromatin immunoprecipitation (ChIP) and high-resolution tiling arrays. Cohesin covered the *GAL2* region in the absence of T7 RNAP (Fig EV4A) but was cleared from this region in a strain that expressed T7 RNAP (Fig EV4B) and accumulated at the site of a T7 terminator sequence in a strain in which this sequence has been inserted downstream of the *GAL2* gene (Fig EV4C). This suggests that T7 RNAP expression in budding yeast can relocate cohesin *in vivo*, indicating that T7 RNAP represents a valid model for analyzing transcriptional effects on cohesin.

In the presence of nucleoside triphosphates (NTPs), recombinant $^{Halo-TMR}$T7 RNAP (Appendix Fig S4G and H) rapidly bound to pPlat DNA into which we had inserted a 20-bp T7 RNAP promoter sequence (pPlat-T7) and translocated uni-directionally at a rate similar to published estimates (Fig 4A; Zhang *et al*, 2014). Multiple fluorescent $^{Halo-TMR}$T7 RNAP structures were seen per DNA. Many of these moved over distances > 10 kb. Translocation was halted upon NTP washout and restarted following their re-addition (Fig 4B and C), indicating that $^{Halo-TMR}$T7 RNAP movement represents transcription.

To test whether transcription can displace cohesin, we stalled $^{Halo-TMR}$T7 RNAP, flowed in cohesin and then restarted transcription. $^{Halo-TMR}$T7 RNAP was able to displace cohesin and translocate it over several kb (Fig 4D). Since Scc1$^{GFP-TEV}$-cohesin and $^{Halo-TMR}$T7 RNAP often co-localized even when not moving, implying that they might interact with each other, we could not determine whether $^{Halo-TMR}$T7 RNAP pushed or pulled cohesin.

The above experiments were performed in a low-salt buffer because T7 RNAP does not transcribe DNA in the presence of higher salt concentrations. To determine whether $^{Halo-TMR}$T7 RNAP could also displace salt-resistant cohesin, that is, cohesin that might entrap DNA, we first washed DNA-bound cohesin with medium-salt buffer and then added $^{Halo-TMR}$T7 RNAP and NTPs in the same buffer conditions as in Fig 4A and D. Under these conditions, $^{Halo-TMR}$T7 RNAP diffused rapidly along the DNA with cohesin. Occasionally, cohesin-$^{Halo-TMR}$T7 RNAP complexes converted to unidirectional movement, suggesting either that $^{Halo-TMR}$T7 RNAP could transcribe when bound to cohesin that might have entrapped DNA, or that these complexes could be displaced by other transcribing $^{Halo-TMR}$T7 RNAP molecules (Fig 4E). Even though cohesin does not normally encounter bacteriophage enzymes, these results indicate that cohesin can be constrained in its movement and be displaced by transcription and imply that similar movements may be mediated by eukaryotic RNA polymerases.

If cohesin translocates on DNA in a manner that is constrained by transcription, it is conceivable that cohesin accumulates at CTCF sites *in vivo* because CTCF acts as a physical barrier to cohesin movement. To test this, we first generated Halo-tagged CTCF (Appendix Fig S7A) and characterized its DNA binding activity using electrophoretic mobility shift assays. $^{Halo-TMR}$CTCF bound to a radiolabeled 100-bp DNA probe containing a CTCF-binding site from the H19/IGF2 imprinted control region (m3 wt) and to a probe containing a "high occupancy" CTCF-binding site (High Oc1, Plasschaert *et al*, 2014) but not to a probe containing a mutated m3 sequence (m3 mt, Ishihara *et al*, 2006; Appendix Fig S7B). Binding of $^{Halo-TMR}$CTCF to m3 wt DNA could be outcompeted using an excess of unlabeled wild-type but not mutated DNA (Appendix Fig S7C). When $^{Halo-TMR}$CTCF was introduced into a flow cell containing 26,323-bp DNA molecules with an array of four closely spaced high occupancy CTCF sites (pPlat 4xCTCF) and exposed to a medium-salt wash, some $^{Halo-TMR}$CTCF molecules were observed at various positions on the DNA or translocated dynamically, but in most cases, CTCF was enriched at the position containing the array of CTCF-binding sites (Fig 5A), indicating that $^{Halo-TMR}$CTCF can recognize its cognate binding sites *in vitro*.

**Figure 3. Cohesin bypasses DNA-bound TetR$^{Halo-TMR}$, $^{Halo-TMR}$EcoRI$^{E111Q}$, and TMR-labeled nucleosomes but not $^{QDot}$EcoRI$^{E111Q}$.**

A–C    Kymograph of salt-resistant Scc1$^{GFP-TEV}$-cohesin diffusing past (A) TetO-DNA-bound TetR$^{Halo-TMR}$, (B) DNA-bound $^{Halo-TMR}$EcoRI$^{E111Q}$, and (C) a pPlat-DNA-bound TMR-labeled nucleosome.
D    Kymograph of salt-resistant Scc1$^{GFP-TEV}$-cohesin failing to bypass DNA-bound $^{QDot}$EcoRI$^{E111Q}$. DNA was stained with Sytox Orange.

Data information: Scale bar = 5 μm.

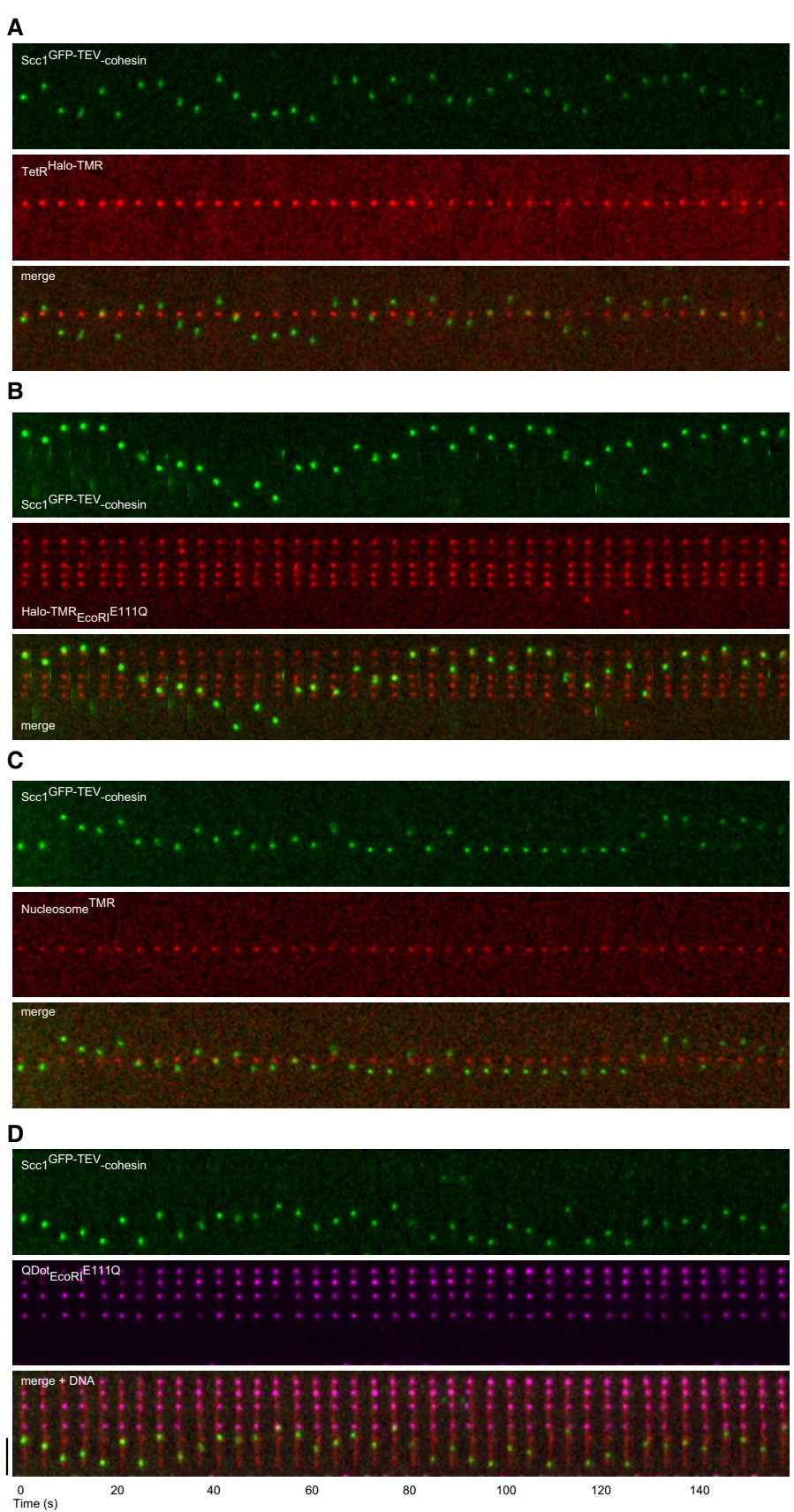

**Figure 3.**

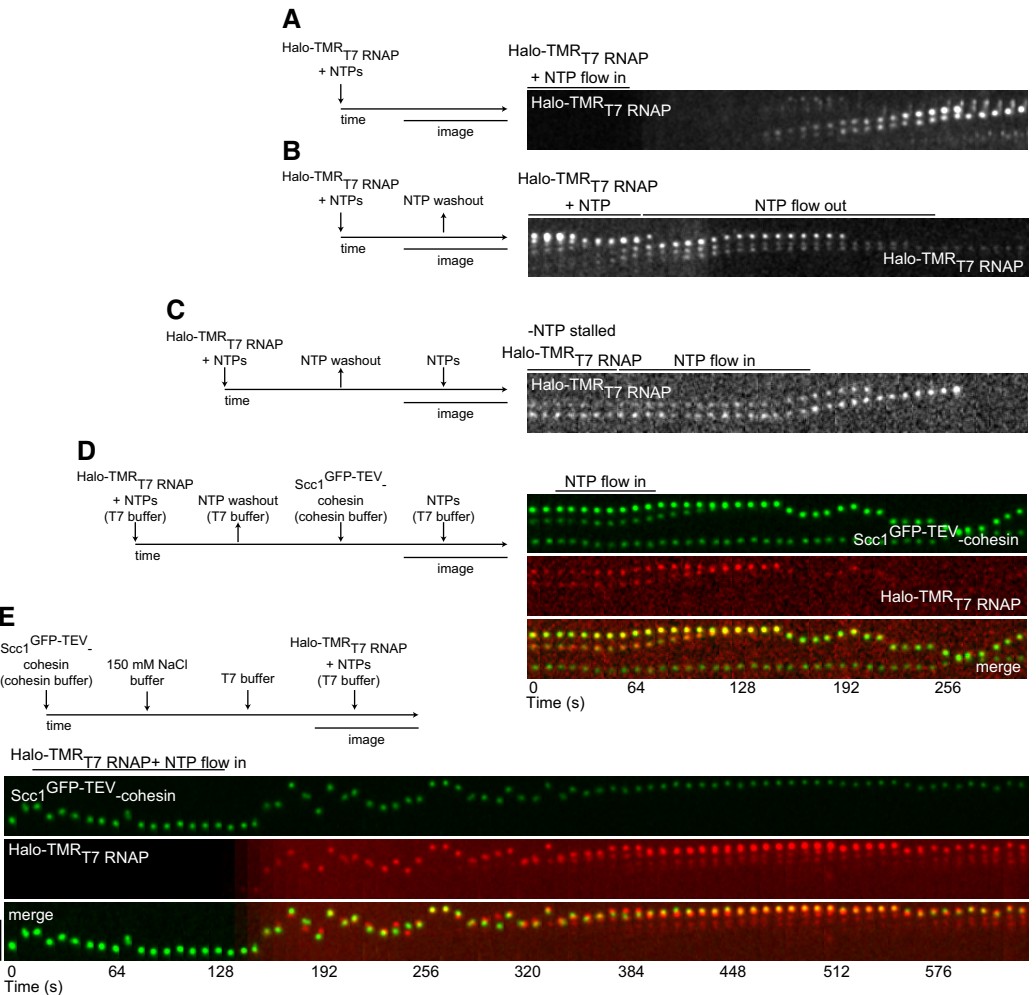

**Figure 4. Transcription constrains cohesin translocation.**

A    Kymograph of two $^{Halo-TMR}$T7 RNAP transcription elongation events.

B, C    Kymographs showing (B) $^{Halo-TMR}$T7 RNAP transcription stalling following removal of NTPs and (C) subsequent resumption after NTP flow in.

D    Kymograph of $^{Halo-TMR}$T7 RNAP (diameter ~8 nm excluding HaloTag) displacing Scc1$^{GFP-TEV}$-cohesin following resumption of transcription in T7 reaction buffer + NTPs.

E    Kymograph of $^{Halo-TMR}$T7 RNAP constraining translocation of salt-resistant Scc1$^{GFP-TEV}$-cohesin.

Data information: Flow in from top and scale bar = 5 μm.

To test whether CTCF constrains the movement of cohesin, we bound Halo-tagged CTCF to pPlat-4xCTCF, flowed in Scc1$^{GFP-TEV}$-cohesin and washed with medium-salt buffer. Out of 40 instances in which Scc1$^{GFP-TEV}$-cohesin translocated independently of CTCF molecules that were immobilized at the expected position on the DNA template, 34 failed to pass $^{Halo-TMR}$CTCF in 160 s and instead translocated away from $^{Halo-TMR}$CTCF again (Fig 5B and Appendix Fig S7D and E). Similar results were obtained using a DNA template containing a single CTCF-binding site (41 out of 56 cohesin structures failed to pass CTCF, Appendix Fig S7F). This indicates that CTCF *per se* accounts for this effect, rather than changes in DNA topology that could occur as the result of interactions between CTCF molecules associated with different binding sites. DNA-bound CTCF therefore poses an obstacle to translocating cohesin and may thus contribute to positioning of cohesin in the genome by functioning as a boundary element.

## Discussion

Although cohesin could principally mediate cohesion by connecting sister chromatids at any position, the genomic distribution of cohesin as analyzed by chromatin immunoprecipitation (ChIP) techniques is highly uneven in species from yeast to men (Blat & Kleckner, 1999; Megee *et al*, 1999; Tanaka *et al*, 1999; Parelho *et al*, 2008; Wendt *et al*, 2008). Cohesin is most enriched at centromeres where it is thought to confer particularly strong cohesion that can resist the pulling force of spindle microtubules (Megee *et al*, 1999; Tanaka *et al*, 1999). Cohesin accumulation at particular

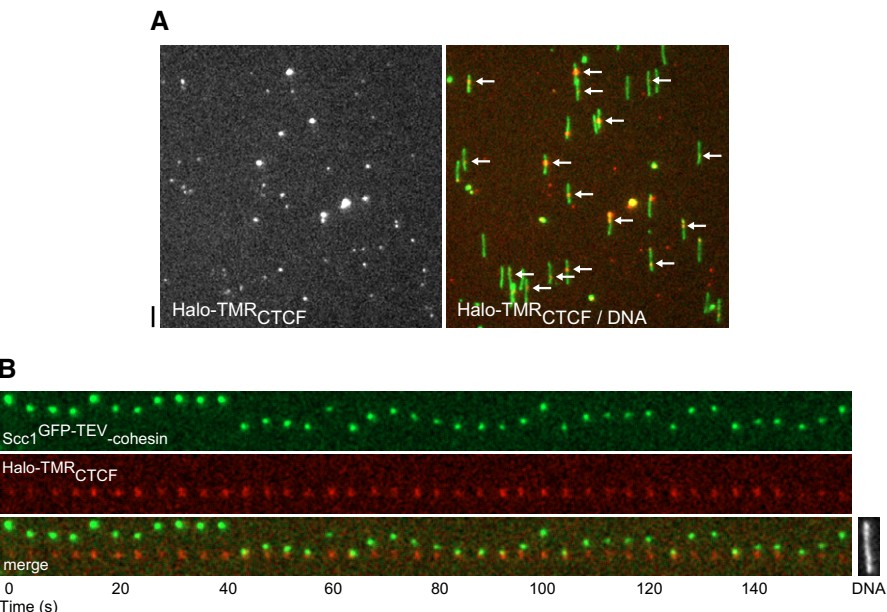

**Figure 5. CTCF constrains cohesin translocation.**

A  Representative field of view showing <sup>Halo-TMR</sup>CTCF bound to pPlat-4xCTCF following 75 mM NaCl, 75 mM KCl buffer wash. Arrows denote CTCF bound at predicted site of 4xCTCF array.

B  Kymograph of salt-resistant Scc1<sup>GFP-TEV</sup>-cohesin failing to bypass DNA-bound <sup>Halo-TMR</sup>CTCF. DNA was post-stained with Sytox Green.

Data information: Scale bars = 5 μm.

chromosomal arm sites is also thought to contribute to cohesion, but has in addition been functionally attributed to the formation of long-range chromosomal *cis* interactions in *Drosophila* and mammalian genomes (Rollins *et al*, 1999; Hadjur *et al*, 2009; Nativio *et al*, 2009; Kagey *et al*, 2010; Guo *et al*, 2011; Seitan *et al*, 2011; Medvedovic *et al*, 2013; Seitan *et al*, 2013). Despite the static picture of cohesin distribution seen in ChIP experiments, it has long been suspected that the distribution of cohesin on DNA must be dynamic, as there is evidence in yeast that cohesin loaded onto DNA at centromeres is relocated to chromosomal arm sites (Megee *et al*, 1999; Hu *et al*, 2011) and that transcription can relocate cohesin (Glynn *et al*, 2004; Lengronne *et al*, 2004; Schmidt *et al*, 2009; Ocampo-Hafalla *et al*, 2016). As in yeast, the mammalian cohesin loading complex has been detected at genomic sites that are distinct from sites at which cohesin accumulates (Kagey *et al*, 2010; Zuin *et al*, 2014b), consistent with the possibility that cohesin is also recruited to DNA by the loading complex at specific sites and subsequently positioned elsewhere. However, the mechanistic basis of these re-localization processes remained poorly understood. It has been proposed that cohesin can slide along DNA (Lengronne *et al*, 2004), but it is also conceivable that cohesin would be positioned at distant sites by the cohesin loading complex *in trans*, which could contact such sites via chromatin looping (discussed in Peters & Nishiyama, 2012).

Our work and the recent observations by Stigler *et al* (2016) reveal that cohesin can indeed translocate along DNA by passive diffusion at a rate of $1.72 \pm 0.1\ \mu m^2/s$ (our study) – $3.8 \pm 0.2\ \mu m^2/s$ (Stigler *et al*, 2016). This is comparable to the diffusion coefficient of soluble cohesin in cells, which has been

estimated to be $2.96 \pm 0.19\ \mu m^2/s$ (Ladurner *et al*, 2014). This phenomenon is difficult to explain by cohesin–DNA interaction modes other than the embracement model, according to which cohesin would entrap DNA inside its ring structure (Haering *et al*, 2008). Furthermore, we found that cohesin–DNA interactions in our *in vitro* assays were abrogated by either DNA or cohesin cleavage but resistant to high-salt (750 mM NaCl) treatment, properties that reflect how cohesin interacts with DNA in cells (Ciosk *et al*, 2000; Gruber *et al*, 2003; Ivanov & Nasmyth, 2005) and which strongly support the embracement model. As predicted by this model, we found that a cohesin subcomplex only containing Smc1 and Smc3, which cannot form stably closed ring structures, was unable to associate with DNA under our assays conditions, further indicating that the cohesin–DNA interactions we observed were not simply caused by non-specific binding modes. Surprisingly, however, cohesin could associate with DNA in the absence of the cohesin loading complex and ATP, both of which are thought to be essential for cohesin loading onto DNA *in vivo* (Ciosk *et al*, 2000; Arumugam *et al*, 2003; Weitzer *et al*, 2003; Hu *et al*, 2011; Ladurner *et al*, 2014). Interestingly, it has recently been observed that a specific cohesin mutant that is defective in ATPase activity can be loaded onto DNA and mediate cohesion *in vivo* (Camdere *et al*, 2015; Elbatsh *et al*, 2016), and a low level of *in vitro* loading of fission yeast cohesin onto DNA in the absence of the loading complex and ATP has also been observed (Murayama & Uhlmann, 2014). We therefore suspect that under our assay conditions cohesin complexes can open and close spontaneously to entrap DNA. If correct, the cohesin loading complex and ATP might have catalytic roles in cohesin loading, that is, they would affect the rate and

required activation energy but not the outcome of the cohesin loading process.

How single cohesin complexes bind to and diffuse along fully chromatinized DNA at physiological salt concentrations remains an important question for the future, but the observation made by both Stigler *et al* (2016) and us that cohesin can pass nucleosomes and several different DNA-bound proteins implies that cohesin may also be able to translocate along chromatin in cells. Interestingly, both human (our study) and fission yeast cohesin (Stigler *et al*, 2016) could pass DNA-bound proteins smaller than ~11 nm in diameter but not obstacles with a diameter of more than ~21 nm. These findings imply that cohesin may not exist in the "open" conformation with an inner ring diameter of 35 nm in which cohesin has been observed by rotary shadowing electron microscopy when it is not associated with DNA (Anderson *et al*, 2002; Huis in 't Veld *et al*, 2014). Instead, cohesin may adopt a different conformation on DNA in which the ring diameter is smaller. Conformations other than the open ring state have indeed been observed for condensin, *Bacillus subtilis* SMC and recently also for human cohesin complexes, but in all these cases the coiled coil regions of the SMC proteins were "closed", creating rod-shaped complexes with little central opening at all (Melby *et al*, 1998; Anderson *et al*, 2002; Huis in 't Veld *et al*, 2014; Soh *et al*, 2015; Hons *et al*, 2016). The finding that cohesin can pass obstacles up to 11 nm in diameter implies that cohesin embracing DNA may exist in yet a different conformation in which its ring structure is partially opened or can rapidly alternate between different conformations.

Interestingly, we found that cohesin translocates past nucleosomes more readily than past [Halo]CTCF, even though a nucleosome (molecular mass 110 kDa, diameter ~11 nm) might be a physically larger obstacle than a single DNA-bound [Halo]CTCF molecule (CTCF has a molecular mass of 83 kDa, but its atomic structure and precise diameter are not known; in addition, the CTCF used in our experiments was fused to HaloTag with a mass of 33 kDa). Since CTCF constrains cohesin translocation irrespective of whether the DNA template contains one or four CTCF-binding sites, DNA looping between CTCF molecules is unlikely to account for this effect. Nevertheless, a change in DNA conformation caused by CTCF binding at a single site cannot be excluded. Alternatively, CTCF's reported ability to multimerize *in vitro* (Pant *et al*, 2004; Yusufzai *et al*, 2004; Bonchuk *et al*, 2015) could account for its ability to prevent cohesin from passing. In either case, our results indicate that CTCF can function as a boundary for translocating cohesin, a phenomenon that may contribute to the accumulation of cohesin at CTCF sites *in vivo*. As the diameter of T7 RNAP (~8 nm) fused to HaloTag (~4 nm) is expected to be smaller than the diameter of cohesin, it was also surprising to find that T7 RNAP could constrain cohesin movements *in vitro* and *in vivo* (this study and Ocampo-Hafalla *et al*, 2016). This raises the possibility that an additional process, perhaps DNA melting or RNA transcription, is responsible for constraining cohesin translocation. Experiments in which the transcriptional activity of T7 RNAP is inhibited using T7 lysozyme could be informative in this regard.

Our finding that cohesin can translocate rapidly along DNA, and does so in a uni-directional manner if encountering a transcribing polymerase, provides a potential mechanistic explanation for how cohesin might be translocated from loading sites to other genomic loci, such as CTCF sites (Lengronne *et al*, 2004; Kagey *et al*, 2010;

Hu *et al*, 2011; Zuin *et al*, 2014a). Cohesin's movability may also be important to allow the unhindered translocation of RNA polymerases and other enzymes along DNA without having to release and reload cohesin, a process that would destroy cohesin-mediated chromosomal interactions. The latter would be particularly harmful for post-replicative cells since cohesin cannot establish cohesion again once DNA replication has been completed (Tachibana-Konwalski *et al*, 2010). Our observation that transcription can provide directionality to cohesin movements also provides a possible explanation for how cohesin could generate long-range chromosomal *cis* interactions via a hypothetical loop extrusion mechanism (Nasmyth, 2001; Nichols & Corces, 2015; Sanborn *et al*, 2015; Fudenberg *et al*, 2016). Since large parts of the genome are transcribed, cohesin could be translocated over long genomic regions until it encounters CTCF at its cognate binding sites or is released by Wapl.

## Materials and Methods

### Protein expression and purification

Scc1[GFP] and Scc1[GFP-TEV]-human cohesin baculoviruses for protein expression in Sf9 insect cells were generated as described (Huis in 't Veld *et al*, 2014). To generate Scc1[Halo-TEV] and Smc3[Halo-Flag], the HaloTag open reading frame (ORF) was PCR amplified from pH6HTN (Promega) and inserted into a vector containing an Scc1[TEV] or Smc3 insect cell expression cassette using Gibson assembly (New England Biolabs Inc). Expression cassettes were combined on multi-gene plasmids using biGBac (Weissmann *et al*, 2016) to generate Smc1[10xHis]/Smc3[Halo-Flag] dimeric cohesin and Smc1 (wt or K38A)/Smc3[Flag] (wt or K38A)/Scc1[Halo-TEV]/[10xHis]SA1 tetrameric cohesin plasmids. Tetrameric cohesin complexes were expressed in Sf9 insect cells and purified via Ni-NTA agarose (Qiagen) followed by anti-FLAG M2 agarose (Sigma-Aldrich) as described (Huis in 't Veld *et al*, 2014) except that the complexes used in Figs EV2 and EV3 were eluted in 50 mM HEPES pH 7.5, 150 mM NaCl, 2 mM MgCl$_2$, 5% glycerol, 0.5 mg/ml Flag peptide. Dimeric cohesin was expressed and purified identically, except a single-step Flag purification was performed. HaloTag Tetramethylrhodamine (TMR) ligand (Promega) was diluted in anti-FLAG binding buffer, incubated for 15 min at room temperature with Scc1[Halo-TEV]-cohesin while immobilized on anti-FLAG M2 agarose, and then washed extensively with anti-FLAG binding buffer.

To generate TetR[Halo-10xHis], the TetR and HaloTag-10xHis ORFs were PCR amplified and combined with pET21a (Merck Millipore) using Gibson assembly. pET21a TetR[Halo-10xHis] was transformed into BL21 (DE3) RIL *Escherichia coli*. Cultures were grown in Lysogeny Broth (LB) supplemented with appropriate antibiotics and expression was induced at mid-logarithmic growth phase with 0.4 mM isopropyl β-D-1-thiogalactopyranoside (IPTG) overnight at 16°C. Cell pellets were resuspended in TetR purification buffer (50 mM Na$_2$HPO$_4$, 300 mM NaCl, pH 8.0), supplemented with 10 mM imidazole, 1 mM DTT, 0.05% Tween-20, and complete protease inhibitor–EDTA (Roche). After sonication and centrifugation at 48,000 *g* for 30 min at 4°C, the soluble fraction was incubated with NiNTA agarose for 45 min and washed with TetR purification buffer supplemented with 20 mM imidazole. HaloTag TMR ligand was incubated with TetR[Halo-10xHis] for 15 min at room temperature while bound to NiNTA

agarose. Protein was eluted with TetR purification buffer supplemented with 250 mM imidazole. TetR[Halo-TMR-10xHis] containing fractions were further purified over a Superdex 200 column (GE Healthcare Life Sciences) in 25 mM Tris pH 7.5, 100 mM NaCl, 5 mM $MgCl_2$, 1 mM DTT, 1 mM EDTA.

To generate [6xHis-Halo]EcoRI[E111Q], the EcoRI[E111Q] ORF was PCR amplified from pEQ111m (Wright *et al*, 1989) and combined with 6xHis-HaloTag and pET21a using Gibson assembly. pET21a [6xHis-Halo]EcoRI[E111Q] was transformed into BL21 (DE3) Rosetta 2 pLysS *E. coli* (Merck Millipore). Expression cultures were grown as described above and induced with 0.5 mM IPTG for 6 h at 37°C, and purified as described (Graham *et al*, 2014). HaloTag TMR or Alexa488 ligand was incubated with [6xHis-Halo]EcoRI[E111Q] for 15 min at room temperature while bound to NiNTA agarose.

To generate [3xMyc-6xHis-Halo]T7 RNAP, the T7 RNAP ORF was PCR amplified from pBioT7 (Eriksen *et al*, 2013) and combined with 3xMyc-6xHis-HaloTag and pBAD (ThermoFisher Inc.) using Gibson assembly. pBAD [3xMyc-6xHis-Halo]T7 RNAP was transformed into Top10 *E. coli*. Expression cultures were grown as described above and induced with 0.2 g/l L-arabinose for 4 h at 37°C. Cell pellets were resuspended in T7 purification buffer (50 mM Tris pH 8.0, 300 mM NaCl, 10% glycerol, 5 mM β-mercaptoethanol) supplemented with 10 mM imidazole, 0.1% Tween-20, 0.1 mM PMSF, and complete protease inhibitor–EDTA (Roche)) and incubated with lysozyme (1 mg/ml cell suspension) for 30 min at 4°C. After sonication and centrifugation at 48,000 *g* for 30 min at 4°C, the soluble fraction was incubated with NiNTA agarose for 90 min, washed with T7 purification buffer supplemented with 10 mM imidazole, 0.01% Tween-20, 0.1 mM PMSF, and then with T7 purification buffer supplemented with 20 mM imidazole and 0.01% Tween-20. HaloTag TMR ligand was then incubated with NiNTA-bound [3xMyc-6xHis-Halo]T7 RNAP for 15 min at room temperature. Protein was eluted with T7 purification buffer supplemented with 300 mM imidazole and 0.01% Tween-20 and dialyzed overnight against 20 mM potassium phosphate buffer pH 7.5, 100 mM NaCl, 10 mM DTT, 0.1 mM EDTA, 50% glycerol.

To generate [10xHis-Halo]CTCF, human CTCF cDNA was PCR amplified from pFastBac HTc CTCF and combined with HaloTag cDNA in a baculovirus expression plasmid under the control of a polyhedrin promoter. [10xHis-Halo]CTCF baculovirus for protein expression in Sf9 insect cells was generated as described (Huis in 't Veld *et al*, 2014). Baculovirus-infected cell pellets were lysed by Dounce homogenization and resuspended in CTCF purification buffer (25 mM HEPES–KOH pH 8.3, 200 mM NaCl, 150 mM KCl, 100 μM $ZnCl_2$, 5% glycerol), supplemented with 10 mM imidazole, 1 mM DTT, 0.05% Tween-20, 1 mM PMSF, and complete protease inhibitor–EDTA (Roche). After centrifugation at 48,000 *g* for 1 h at 4°C, the soluble fraction was incubated with NiNTA agarose for 45 min and washed with CTCF purification buffer supplemented with 20 mM imidazole. HaloTag TMR ligand was incubated with [10xHis-Halo]CTCF for 15 min at room temperature while bound to NiNTA agarose. Protein was eluted with CTCF purification buffer supplemented with 250 mM imidazole and 1 mM DTT and dialyzed against CTCF purification buffer supplemented with 1 mM DTT for 2.5 h at 4°C.

For QDot conjugation to EcoRI[E111Q], 3xMyc-6xHis was added at the N-terminus of EcoRI[E111Q] using Gibson assembly and expressed and purified as above. Anti-myc antibody 4A6 (Millipore) was labeled with QDot 705 (SiteClick Qdot 705 Antibody Labeling Kit; ThermoFisher Scientific) and mixed with [3xMyc-6xHis]EcoRI[E111Q] prior to incubation with λ-DNA.

## Histone octamer expression and nucleosome reconstitution

Amino acid substitutions in *Xenopus laevis* histone H3 (C110A, Q125C) and histone H2B (K113C) were introduced in the polycistronic plasmid pET29a-YS14 by site-directed mutagenesis. Recombinant histone octamers were expressed in *E. coli* and purified under native conditions as described (Shim *et al*, 2012) except that following NiNTA purification the peak histone-containing fractions were incubated with a 40-fold molar excess of tetramethylrhodamine-5-maleimide (Sigma-Aldrich) overnight at 4°C. Unreacted dye was quenched with DTT and separated using Sephadex G50 Fine (GE Healthcare Life Sciences). Histone octamers were purified over a Superdex 200 column (GE Healthcare Life Sciences) and were then mixed with 1 μg of biotinylated pPlat or pPlat-601 at a molar ratio of ~80 octamer:1 DNA in 10 mM Tris pH 7.6, 2 M NaCl, 1 mM EDTA, and incubated for 30 min at 4°C. Histone octamers were deposited on DNA by stepwise dilution at 4°C to reduce of NaCl concentration at 4°C (1 h at 1 M, 1 h at 0.8 M, 1 h at 0.67 M, 1 h at 0.2 M, overnight at 0.1 M). Reconstituted nucleosomal DNA was stored at 4°C.

## Single-molecule cohesin: DNA binding assay

Biotinylated polyethylene glycol functionalized coverslips (MicroSurfaces Inc.) were assembled into flow chambers (Yardimci *et al*, 2010, 2012). Flow chambers were incubated with 1 mg/ml Avidin DN (Vector Laboratories) for 30 min and washed with 20 mM Tris pH 7.5, 50 mM NaCl, 2 mM EDTA, 0.1 mg/ml UltraPure BSA (ThermoFisher Scientific) using a syringe pump at a flow rate of 50 μl/min. 0.5 ml of the above buffer supplemented with biotinylated λ genomic DNA (2.3 pM final concentration) or 0.5 ml biotinylated pPlat (1.7 pM final concentration) (see below) was introduced at 50 μl/min. Following washout of unbound DNA molecules, cohesin was flowed in at 3–5 nM in experiments presented in Figs 1E and F, and EV1D–H, and Appendix Fig S3 and at 0.7 nM in all other experiments. Cohesin was flowed in cohesin binding buffer (35 mM Tris, pH 7.5, 25 mM NaCl, 25 mM KCl, 1 mM $MgCl_2$, 10% glycerol (v:v), 0.003% Tween-20, and 0.1 mg/ml UltraPure BSA) at 20 μl/min and incubated for 5–10 min. Flow chambers were then washed with the same buffer, and optionally with 750 mM NaCl buffer (35 mM Tris, pH 7.5, 750 mM NaCl, 10 mM EDTA, 10% glycerol, 0.35% Triton X-100, and 0.1 mg/ml UltraPure BSA) to dissociate non-topologically bound cohesin complexes. Sytox Orange or Sytox Green DNA stain (ThermoFisher Scientific) was included in imaging buffers in experiments that required visualization of DNA (~5 nM–500 nM, depending on imaging buffer salt concentration). Single-molecule imaging experiments were performed at room temperature (~23°C).

A glucose oxidase/catalase/glucose oxygen scavenger system was included in all imaging buffers in experiments described in Figs 1E, 2 and EV1D, F, H, and Appendix Fig S3 [final concentration 4.5 mg/ml glucose, 0.2 mg/ml glucose oxidase, 35 μg/ml catalase, 1% betamercaptoethanol (Sigma-Aldrich)]. A protocatechuic acid (PCA)/protocatechuate-3,4-dioxygenase (PCD)/Trolox oxygen

scavenger system was included in all imaging buffers in the experiments described in Figs 3–5 and Appendix Figs S4B, D, F, and S7D–F (final concentration 10 nM PCD, 2.5 mM PCA, 2 mM Trolox; Sigma-Aldrich; Aitken *et al*, 2008).

For XhoI λ-DNA restriction digest experiments, cohesin was flowed in as described above. The flow chamber was washed with 750 mM NaCl buffer and then with XhoI digestion buffer (10 mM HEPES–KOH pH 7.7, 200 mM KCl, 10 mM MgCl$_2$, 0.35% Triton X-100, 0.1 mg/ml UltraPure BSA (ThermoFisher Scientific)). FastDigest XhoI (ThermoFisher Scientific) was then flowed in during imaging.

For the cohesin cleavage experiments described in Fig EV3A–C, Smc1, Smc3$^{FLAG}$, Scc1$^{Halo-TMR-TEV}$, $^{10xHis}$SA1 cohesin was incubated ± TEV protease (generated in-house) during purification while immobilized on anti-FLAG M2 agarose (3 h, 4°C) and eluted in 50 mM HEPES pH 7.5, 150 mM NaCl, 2 mM MgCl$_2$, 5% glycerol, 0.5 mg/ml FLAG peptide. For TEV protease flow in experiments, cohesin was flowed in and the flow chamber was washed with 750 mM NaCl buffer supplemented with 2 mM DTT; TEV protease was then flowed in during imaging.

For TetR$^{Halo-TMR}$ experiments, pPlat-TetO DNA flow chambers were washed with cohesin binding buffer. TetR$^{Halo-TMR}$ was flowed in at 7 nM in cohesin binding buffer, incubated for 4 min, and then washed with cohesin binding buffer. Cohesin was flowed in as described above, washed with cohesin binding buffer and then with cohesin binding buffer in which the salt concentration was increased to 75 mM NaCl, 75 mM KCl.

For $^{Halo-TMR}$EcoRI$^{E111Q}$ and $^{Halo-A488}$EcoRI$^{E111Q}$ experiments, λ-DNA (112 pM) was incubated with $^{Halo}$EcoRI$^{E111Q}$ (5 nM dimer) in 20 μl EcoRI buffer (20 mM Tris pH 7.5, 150 mM KCl, 0.1 mg/ml BSA) for 30 min at room temperature. The reaction was then diluted to 0.5 ml with EcoRI buffer and drawn into the flow chamber. Non-specifically bound $^{Halo}$EcoRI$^{E111Q}$ was washed out with EcoRI buffer. Buffer was exchanged with cohesin binding buffer prior to cohesin flow in. Cohesin was flowed in as described for the TetR$^{Halo-TMR}$ experiments. $^{QDot}$EcoRI$^{E111Q}$ experiments were performed identically, except that $^{3xMyc-6xHis}$EcoRI$^{E111Q}$ was pre-incubated with 0.3 μl anti-Myc QDot 705 prior to incubation with λ-DNA.

For nucleosome experiments, nucleosomal-pPlat was drawn into the flow chamber and washed with cohesin binding buffer prior to cohesin flow in. Cohesin was flowed in as described for the TetR$^{Halo-TMR}$ experiments except cohesin was washed with cohesin binding buffer in which the salt concentration was increased to 150 mM NaCl, 150 mM KCl.

For $^{Halo-TMR}$T7 RNAP experiments, pPlat-T7 DNA flow chambers were washed with T7 reaction buffer (40 mM Tris pH 7.9, 6 mM MgCl$_2$, 5 mM DTT) prior to polymerase flow in. $^{Halo-TMR}$T7 RNAP was flowed in at 5 nM in 50 μl T7 reaction buffer supplemented with 2 mM NTPs, 1.25 μl RNase OUT (ThermoFisher Scientific), and 1× PCA/PCD/Trolox oxygen scavenger mix.

For $^{Halo-TMR}$CTCF experiments, pPlat-4xCTCF or pPlat-1xCTCF DNA flow chambers were washed with cohesin binding buffer. $^{Halo-TMR}$CTCF was flowed in at 0.3 nM in cohesin binding buffer, incubated for 10 min, and then washed with cohesin binding buffer in which the salt concentration was increased to 75 mM NaCl, 75 mM KCl. Buffer was exchanged with cohesin binding buffer prior to cohesin flow in. Cohesin was flowed in as described for the TetR$^{Halo-TMR}$ experiments.

## DNA templates for single-molecule imaging

Doubly biotinylated bacteriophage λ genomic DNA was prepared as described (Yardimci *et al*, 2012), except that Taq DNA ligase (New England Biolabs Inc.) was used instead of T4 DNA ligase. To generate pPlat-TetO and pPlat-T7, the plasmid pPlat (25,754 bp) was linearized with FspAI and a PCR product containing seven copies of the TetO sequence amplified from pTRE3G (Clontech Laboratories Inc.) or a PCR product containing the T7 promoter and a 1.5-kb yeast genomic DNA sequence amplified from plasmid pFL2_CasIA were inserted using Gibson assembly. To generate pPlat-601, a DNA fragment containing a single 601 nucleosome positioning sequence (Lowary & Widom, 1998) was generated by primer extension PCR and inserted into pPlat as described above. To generate pPlat-4xCTCF, a cDNA fragment containing four CTCF-binding sites reported to display high affinity CTCF binding (Plasschaert *et al*, 2014) (bGm5, GTCTTCCCTCTAGTGGTAA; 47, CCCGGCGCAGGG GGGCGCTG; 101, CCGGCCGGCAGAGGGCGCGC; 100 mt, CCGGCC AGAAGGGGGCGCGC) each separated by a 100-bp linker was synthesized (Integrated DNA Technologies Inc.) and inserted into pPlat as described above. To generate pPlat-1xCTCF, a DNA fragment containing a single high affinity CTCF-binding site (HighOc1; Plasschaert *et al*, 2014: GCGGCCAGCAGGGGGCGCCC) was generated by primer extension PCR and inserted into pPlat as described above. Doubly biotinylated pPlat DNA was prepared by linearizing pPlat with SpeI and performing PCR extension using biotinylated dATP and dCTP nucleotides and Taq DNA polymerase. Linearized pPlat-TetO is 26,192 bp with TetO at position 10,123–10,561 bp; linearized pPlat-T7 is 27,238 bp with the T7 promoter at position 10,123 bp; linearized pPlat-601 is 25,908 bp with 601 at position 10,123–10,270 bp; linearized pPlat-4xCTCF is 26,323 bp with the 4 CTCF high affinity binding sites at position 10,123–10,692 bp.

## Single-molecule microscopy

Time-lapse microscopy images were acquired using a Zeiss TIRF 3 Axio Observer setup described previously (Mieck *et al*, 2015). Images were acquired at 4-s intervals unless otherwise stated. High temporal resolution single-molecule imaging (Figs 2, EV2E and EV3B–C, E–F, and Appendix Fig S6; images acquired at 15 Hz) was performed using a custom-built TIRF microscope setup described previously (Mieck *et al*, 2015) and analyzed using software developed in MATLAB (Mathworks Inc.).

## *Xenopus laevis* egg extract preparation and use

*Xenopus laevis* egg extract experiments were performed as described (Huis in 't Veld *et al*, 2014).

## Bulk cohesin: DNA loading assay

Circular nicked (C) pSP64 plasmid (3 kb) was prepared using Nt.BspQI (NEB). Linearized pSP64 (L) was prepared using BamHI (NEB). Plasmids were purified by Qiaquick Gel Extraction (Qiagen). For DNA cleavage experiments, recombinant Smc1, Smc3$^{FLAG}$, Scc1, $^{10xHis}$SA1 cohesin was prepared as described above and eluted in 25 mM Tris pH 7.5, 50 mM NaCl, 50 mM KCl, 2 mM MgCl$_2$, 10% glycerol, 0.5 mg/ml FLAG peptide. For cohesin cleavage

experiments, Smc1, Smc3[FLAG], Scc1[Halo-biotin-TEV], [10xHis]SA1 cohesin was incubated with TEV protease during purification while immobilized on anti-FLAG M2 agarose (3 h, 4°C) and eluted in 35 mM HEPES pH 7.5, 50 mM NaCl, 50 mM KCl, 2 mM MgCl$_2$, 10% glycerol, 0.5 mg/ml FLAG peptide.

Bulk *in vitro* cohesin–DNA loading assay conditions were adapted from Murayama and Uhlmann (2014). For DNA cleavage experiments, cohesin and DNA were combined in a 20-µl reaction (final composition: 45 nM cohesin, 3.3 nM DNA, 35 mM Tris pH 7.5, 56 mM NaCl, 19 mM KCl, 1 mM MgCl$_2$, 1 mM TCEP, 10% glycerol, 0.003% Triton X-100) and incubated for 1 h at 32°C. Stop buffer (180 µl; 35 mM Tris pH 7.5, 500 mM NaCl, 20 mM EDTA, 0.35% Triton X-100, 5% glycerol) was added for 5 min at 32°C to dissociate non-topologically bound cohesin from DNA. Reactions were then combined with 15 µl anti-Scc1 (A900) (Waizenegger *et al*, 2000) coupled Affi-Prep protein A resin (Bio-Rad), diluted to 400 µl with low-salt buffer (35 mM Tris pH 7.5, 150 mM NaCl, 0.35% Triton X-100, 5% glycerol), and incubated for 2 h at 4°C. Resin was washed twice with high-salt buffer (35 mM Tris pH 7.5, 750 mM NaCl, 0.35% Triton X-100, 5% glycerol) and once with low-salt buffer. Complexes were eluted using Scc1 peptide (30 min, 4°C), digested with proteinase K (2 h, 50°C; 1 mg/ml), and analyzed by 0.8% agarose gel electrophoresis in 1× TAE. DNA was stained using GelRed (Biotium) and imaged using a ChemiDoc XRS+ system (Bio-Rad). Background subtraction was performed in ImageJ. Data from three independent experiments were plotted; error bars denote standard error of the mean.

For cohesin cleavage experiments, cohesin and circular nicked DNA were combined in a 20-µl reaction (final composition: 21 nM cohesin, 3.3 nM DNA, 30 mM HEPES pH 7.5, 56 mM NaCl, 18 mM KCl, 1 mM MgCl$_2$, 0.1 mM TCEP, 5.5% glycerol, 0.003% Triton X-100) and processed as described above. Data from four independent experiments were plotted; error bars denote standard error of the mean.

For cohesin:ATP experiments, cohesin and circular nicked DNA were combined in a 20-µl reaction (final composition: 30 nM cohesin (wt or Smc1/3 K38A), 2.2 nM DNA, 30 mM HEPES pH 7.5, 56 mM NaCl, 18 mM KCl, 1 mM MgCl$_2$, 0.1 mM TCEP, 5.5% glycerol, 0.003% Triton X-100 ± 0.5 mM ATP, ADP, ATP-γS (Jena Bioscience), AMP-PNP (Jena Bioscience) and AMP-PCP (Jena Bioscience)) and incubated for 90 min at 32°C. Reactions were stopped with 300 µl low-salt buffer supplemented with 27 mM EDTA and incubated with 15 µl anti-Scc1 antibody beads and processed as described above. Data from three independent experiments were plotted; error bars denote standard error of the mean.

For the proteinase K only elution experiment described in Fig EV1A, reactions were processed as for the cohesin:ATP experiments, except the beads were incubated with proteinase K (2 h, 50°C; 0.5 mg/ml) directly following high-salt and low-salt washes. Data from three independent experiments were plotted; error bars denote standard error of the mean.

## ATPase assay

Cohesin complexes were incubated in 50 mM HEPES pH 7.5, 150 mM NaCl, 1 mM MgCl$_2$, 5% glycerol, 0.1 mg/ml BSA, 10 nM [γ-$^{32}$P]-ATP, and 50 µM non-radiolabeled ATP. Reactions were incubated at 32°C and stopped by adding 1% SDS and 10 mM EDTA.

Reaction products were separated on polyethyleneimine plates (EMD Biosciences) by thin-layer chromatography using 0.75 M KH$_2$PO$_4$ (pH 3.4) and analyzed by phosphor imaging with a Typhoon Trio Scanner (Amersham).

## HeLa Scc1[GFP] cell line generation

Scc1[GFP] HeLa Kyoto cells were generated by CRISPR Cas9-mediated homologous recombination as described (Cong *et al*, 2013). Briefly, cells were transfected with plasmids expressing SpCas9(D10A) nickase and chimeric guide RNAs targeting the region coding for the Scc1 C-terminus and a plasmid that carried the coding sequence for a monomeric version of GFP flanked on either side by 800- to 1,200-bp homology arms. Clonal cell lines were sorted by FACS; recombination and homozygous tagging were assayed by PCR and immunoblotting. Fluorescence microscopy of the endogenous GFP signal from HeLa Kyoto Scc1[GFP] cells was performed using a Zeiss LSM780 Axio Observer confocal microscope.

## Bulk *in vitro* RNA transcription assay

A 119-bp DNA template for *in vitro* transcription containing a T7 promoter was generated by primer extension (oligonucleotides: TAATACGACTCACTATAGTGATAAGTGGAATGCCATGGTTTTAGAG CTAGAAATAGCAAG and AAAAAAGCACCGACTCGGTGCCACTTTT TCAAGTTGATAACGGACTAGCCTTATTTTAACTTGCTATTTCTAGC TCTAAAAC) followed by PCR amplification using oligonucleotides TAATACGACTCACTATA and AAAAAAGCACCGACTCGGTGCCAC. *In vitro* transcription was performed for 4 h at 37°C using [Halo-TMR]T7 RNAP according to NEB T7 RNA Polymerase reaction conditions. The DNA template was removed by adding 1 µl TURBO DNase (Ambion). Nucleic acids were purified by phenol–chloroform extraction and ethanol precipitation and were resuspended in 50 µl RNase-free H$_2$O. 0.2 µl was analyzed using capillary electrophoresis (Fragment Analyzer, Advanced Analytical Technologies Inc.) and the High Sensitivity RNA Analysis Kit (AATI, DNF-472).

## Budding yeast T7 RNAP *in vivo* experiments

Details of the yeast strains and experimental setup have been published elsewhere (Ocampo-Hafalla *et al*, 2016). Cells were processed for chromatin immunoprecipitation as described (Lengronne *et al*, 2004). Pk-tagged Scc1 was immunoprecipitated using anti-Pk antibodies (AbD Serotec, SV5-Pk1) from strains grown in the absence of galactose. Chromatin immunoprecipitates were hybridized to Affymetrix GeneChip *S. cerevisiae* Tiling 1.0R arrays. Enrichment in the immunoprecipitate relative to a whole genome DNA sample is presented. Each bar represents the average of 25 oligonucleotide probes within adjacent 125-bp windows.

## Electrophoretic mobility shift assays

~100-bp dsDNA probes containing a single CTCF-binding site were generated by primer extension PCR. Probe m3 wt contains the 3[rd] CTCF-binding site from the H19/IGF2 ICR region (GGATGCT ACCGCGCGGTGGCAGCA). Probe m3 mt contains a mutated version of m3 (Ishihara *et al*, 2006) (GAAGTTGCCGAGCAGCGACCAGTG). Probe HighOc1 contains a high affinity CTCF-binding site

(Plasschaert *et al*, 2014) (TCAGAGTGGCGGCCAGCAGGGGGCGCC CTTGCCAGA). Probes were labeled with $[\gamma^{32}P]$-ATP using T4 Polynucleotide Kinase (ThermoFisher Scientific).

To compare the binding affinity of $^{Halo}$CTCF to m3 wt, m3 mt, and HighOc1 probes, $^{Halo}$CTCF was combined with the non-specific competitor poly(deoxyinosinic-deoxycytidylic) acid (poly(dI-dC).poly(dI-dC), ThermoFisher Scientific) in a 20-μl reaction for 10 min at room temperature (final composition: 800 fmol $^{Halo}$CTCF, 500 ng poly(dI-dC).poly(dI-dC), 35 mM Tris pH 8.3, 25 mM NaCl, 25 mM KCl, 1 mM MgCl$_2$, 10% glycerol, 1 mM DTT, 0.1 mM ZnCl$_2$). 24 fmol of radiolabeled probe was added and the incubation continued for a further 30 min at room temperature. Reactions were resolved on a 6% polyacrylamide DNA Retardation Gel (Thermo-Fisher Scientific), exposed to a phosphorimager screen, and analyzed using a Typhoon Trio Scanner (Amersham).

To analyze the effect of competitors on the binding affinity of $^{Halo}$CTCF to m3 wt probe DNA, 800 fmol of $^{Halo}$CTCF was combined with 500 ng poly(dI-dC).poly(dI-dC), 7.2 pmol m3 wt, or 7.2 pmol m3 mt for 10 min at room temperature in the above buffer. 24 fmol radiolabeled m3 wt probe was then added, and the reactions were processed as described above.

### Immunoblotting

Previously described antibodies raised against Smc1 (A847) (Sumara *et al*, 2002), Smc3 (A845) (Sumara *et al*, 2002), Scc1 (A900) (Waizenegger *et al*, 2000), and SA1 (A823) (Sumara *et al*, 2000) were used to detect corresponding subunits of recombinant human cohesin. Scc1 (A900) was also used to detect *Xenopus* Scc1. Commercially available antibodies were used to detect GFP (Roche, 11814460001), Scc1 (Appendix Fig S1C, Merck Millipore, 05-908), and H3 (Santa Cruz, sc-8654).

**Expanded View** for this article is available online.

### Acknowledgements

We thank M. Colombini and the IMP mechanical workshop for the manufacturing of equipment used in this study. We are grateful to S. Westermann, C. Mieck, R. Heinen, K. Uzunova, G. Schmauss, T. Lendl, E. Turco, L.D. Gallego, A. Köhler, J.C. Walter, H. Yardimci, and all members of the Peters laboratory for discussions and assistance. We thank the IMP/IMBA BioOptics and Molecular Biology Service for technical assistance. We thank T. Nishiyama for sharing unpublished data. pEQ111m was a gift from P. Modrich (Addgene plasmid # 40190). pET29a-YS14 was a gift from Jung-Hyun Min (Addgene plasmid # 66890). pBio-T7 was a gift from L. Jauffred. Research in the laboratory of J.-M.P. is supported by Boehringer Ingelheim, the Austrian Science Fund (SFB F34 and Wittgenstein award), the Austrian Research Promotion Agency (FFG 852936), the Vienna Science and Technology Fund (WWTF LS09-13), and the European Commission (FP7/2007-2013, no. 241548, MitoSys). M.I.M. acknowledges the VIPS Program of the Austrian Federal Ministry of Science and Research and the City of Vienna. Research in the laboratory of A.V. is supported by the WWTF projects VRG10-11 and LS14-009, the Human Frontiers Science Program Project RGP0041/2012 and Research Institute of Molecular Pathology (IMP). The IMP is funded by Boehringer Ingelheim.

### Author contributions

IFD, DG, MPZ, MIM, PJHV, MO-H, RL, FU, AV, and J-MP designed the experiments and interpreted the data. IFD, DG, MPZ, PJHV, FW, and GL generated

reagents. IFD performed most of the single-molecule imaging experiments and the *Xenopus* experiments. DG carried out the bulk cohesin–DNA loading assays. MPZ performed the $^{Halo-TMR}$CTCF electrophoretic mobility shift assays and the TetR$^{Halo-TMR}$ single-molecule imaging. MPZ and IFD carried out the $^{Halo-TMR}$CTCF single-molecule imaging. MIM and IFD performed the single-molecule imaging described in Fig 2. DAC generated and characterized the HeLa Scc1$^{GFP}$ cell line. MO-H carried out the *in vivo* budding yeast T7 RNAP experiments under the guidance of FU. IFD and J-MP wrote the manuscript.

### Conflict of interest

The authors declare that they have no conflict of interest.

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
