## [Review Process File · The EMBO Journal]

Manuscript EMBO-2016-95402

Rapid movement and transcriptional re-localization of human cohesin on DNA

Iain F Davidson, Daniela Goetz, Maciej P Zaczek, Maxim I Molodtsov, Pim J Huis In 't Veld, Florian Weissmann, Gabriele Litos, David A Cisneros, Maria Ocampo-Hafalla, Rene Ladurner, Frank Uhlmann, Alipasha Vaziri, Jan-Michael Peters

Corresponding author: Jan-Michael Peters, Research Institute of Molecular

Review timeline:

Submission date:	03 August 2016
Editorial Decision:	14 August 2016
Revision received:	08 September 2016
Accepted:	30 September 2016

Editor: Bernd Pulverer

Transaction Report:

Previous Referee Reports – round 1

Referee #1 (Remarks to the Author):

Peters and colleagues describe a TIRF microscopy-based assay to investigate the dynamics of cohesin association with and translocation along DNA. Fluorescently labeled cohesin complexes associate with surface-immobilized DNA under very low (50 mM) salt conditions and remain bound upon increasing salt concentrations (to 750 mM), even in the absence of ATP or the cohesin chromosome loading complex SCC2/4. Notably, cohesin complexes seem to be able to translocate along DNA and thereby pass small (<10 nm) but not large (>20 nm) obstacles bound to the DNA. DNA cleavage or proteolytic cleavage of the cohesin ring release cohesin from the DNA. Cohesin complexes co-migrate on DNA with T7 RNA polymerase and co-localize with CTCF bound to the DNA. These experiments suggest that cohesin rings encircle and slide on (and off the ends of) linear DNAs or chromatin fibers and that certain chromosomal proteins or protein complexes, like CTCF or RNA polymerase complexes, could hinder this movement.

The TIRF assays presented in this (and the accompanying) manuscript is an interesting and potentially powerful approach to study the mechanics of cohesin's action. A number of in vivo and in vitro experiments from the Nasmyth and Uhlmann labs had previously tested the idea that cohesin complexes can encircle and consequently slide along DNA. The data presented in this manuscript lends further support to this hypothesis. While the method presented in this manuscript is without doubt very promising for the development of further studies of cohesin's mechanism, the conceptual advance does, at this point, not go significantly beyond the previous studies. The new experimental system will furthermore require additional controls and a more thorough quantitative analyses to

firmly establish the physiological relevance of the observed cohesin movements on surface-immobilized DNA (see specific comments below). I therefore strongly encourage the authors to consolidate their novel assay system, but think that the work, in its current form, is too preliminary to justify publication in Nature.

Specific comments:

1. The facts that cohesin loading requires extremely low salt concentrations (50 mM), takes place in the absence of cohesin's SCC2/4 loading complex, and is independent of ATP raises strong doubts whether the DNA binding observed *in vitro* properly reconstitutes the loading of cohesin onto chromosomes *in vivo*. This is in contrast to previous *in vitro* DNA binding experiments with fission yeast cohesin, which showed a stimulation in binding by SCC2/4 and ATP (Murayama and Uhlmann, 2014, 2015). One possibility to support the notion that the authors observe a proper association of cohesin holocomplexes with DNA were to test whether, for example, isolated SMC1/3 dimers failed to load onto DNA in the same experimental setup.
2. Under the experimental conditions of the *in vitro* loading assay, most cohesin complexes tend to cluster non-specifically (with each other or with, for example, a bacteriophage RNA polymerase). It is conceivable that the observed behavior might be artificially biased by the clustering of these macromolecules. For example, might the inability of cohesin complexes to pass larger obstacles on the DNA be a consequence of their oligomerization, since one would, in principle, expect that a 50-nm cohesin ring should be able to traverse 20-nm protein barriers (Fig. 3c)? Only "some" cohesin molecules (pg. 6, Fig. 2) allowed real single-molecule analyses. If the authors could find conditions that increased the fraction of single molecule observations, then this could strongly support the conclusion that the assay measures biophysical properties of native cohesin complexes.
3. The manuscript would strongly benefit from more quantitative instead of qualitative analyses. One important control is that TEV cleavage of RAD21 releases cohesin from DNA. However, the only statement in the manuscript is that upon TEV cleavage, "less RAD21-Halo-TMR-TEV cohesin remained DNA-bound." (pg. 5 and Fig. 1f, g). To what degree was the amount of cohesin reduced in the case of the cleavable and non-cleavable RAD21 versions? A quantitative analysis can rule out the impression that also the levels of non-cleavable cohesin are strongly reduced in the presence of the protease (see the image shown in Fig. 1g). Similarly, it would be important to know (see above) which fraction of the cohesin signal corresponds to single cohesin complexes (pg. 6). Comparison of the fluorescence intensities of single cohesin molecules and cohesin clusters might furthermore allow an estimate of the stoichiometry of cohesin in the clusters.
4. In the initial T7 experiments, cohesin clusters seem to translocate with the polymerases (Fig. 4d). This supports the notion that transcription might 'push' (or 'pull') cohesin. However, these experiments have been performed solely under low salt conditions, which suggests that the majority of cohesin complexes observed do not encircle DNA and might rather bind directly (non-specifically) to the RNA polymerases. For the experiments that follow, the authors perform a high salt wash and hence select for cohesin rings that have encircled DNA. Under these conditions, however, only a small fraction of T7 polymerases seem to actively transcribe DNA ("Occasionally, cohesin-Halo-TMR T7 RNAP complexes converted to unidirectional movement", pg. 9 and Fig. 4e). Without further evidence that transcription takes place under these conditions, the conclusion that transcription moves cohesin cannot be made. Is this movement dependent on NTPs? One possible control were to inhibit T7 polymerase (for example by T7 lysozyme) and test whether this prevented unidirectional movement of cohesin- T7 polymerase complexes. Since the authors observe an accumulation of cohesin at T7 terminator sequences *in vivo* (Extended Data Fig. 6c), the authors could also insert the same sequence at a defined position within the DNA template and test whether this results in the accumulation of cohesin at this position *in vitro*.
5. The author suggest that CTCF acts as a specific roadblock for cohesin movement (Fig. 4 g, h). However, it is likely that any protein that would form a large DNA loop would create an obstacle for the passage of a protein that slides along DNA. Since the assay cannot address whether cohesin can support loop formation at CTCF sites, for example by encircling DNA sequences *in cis*, it is not obvious how these experiments add to explaining how a chromatin domain might be formed by CTCF and cohesin *in vivo*. One possibility to test whether CTCF can specifically prevent the translocation of cohesin would be to either replace the 4 CTCF binding sites by a single site,

assuming that this would not result in loop formation, or to use a mutant version of CTCF that cannot multimerize (Bonchuk et al., 2015).

6. The authors claim that the cohesin complexes they use are ring-shaped (pg. 4). Even though they show that the four cohesin subunits assemble into a complex (Fig. 1a), the authors didn't directly test for ring formation. This could, for example, be achieved by testing whether the two fragments of RAD21 generated by TEV cleavage remain associated with an SMC1/3 dimer.

7. To allow readers to judge the efficiency of the co-purification of a "small amount" (pg. 4) of circular DNA with cohesin, the authors should plot the 'fractions of input' in Fig. 1b and d instead of the 'fold change over no cohesin control'.

8. Taking into consideration that the effect of T7 RNA polymerase transcription on the positioning of cohesin on yeast chromosomes has already been described in the supporting manuscript (Ocampo-Hafalla et al.), reproduction of the data shown in Fig. 8b of this manuscript in Extended Fig. 6a and b is redundant.

9. The authors refer to the SCC2/4 cohesin loading complex as CLC. Since the same complex has previously been named 'adherin' or 'kollerin' by different authors, introducing yet another name is very confusing.

Referee #2 (Remarks to the Author):

Using fluorescence imaging techniques, the authors visualize the movement of individual cohesin molecules along DNA. They show that cohesin diffuses along the DNA, that its movement can be blocked by larger structures, and that directionality can be imposed by transcription. These observations are consistent with the picture that the cohesin ring is topologically enclosing DNA and allowing the compaction of genomic DNA by the coupling of diffusional cohesin movement and loop formation. The experiments are very informative, but at this point the data and their interpretation do not convincingly support the main mechanistic conclusions. In particular, the following issues should be addressed:

- The authors show that cohesin compacts singly-tethered DNA and leaves doubly-tethered experiments in a stretched state. For these experiments to be interpreted correctly in the context of incompletely stretched DNA molecules supporting looping by cohesin, the authors should provide information on the actual tether length of the DNA molecules analyzed. This is particularly important when analyzing the behavior of cohesins that seem to be loaded on doubly-tethered DNA: they may correspond to cohesins encircling a single copy of the duplex, or they may encircle two duplexes, holding together a loop. The latter scenario would be consistent with the authors' statement on page 6 that some spots contain multiple copies of cohesin. The simultaneous observation of cohesin signal and a colocalized spot of DNA stain may demonstrate the existence of such loops. Relatedly, the complexes with the highest diffusion coefficient should correspond with spots that contain only one cohesin tetramer; these would be the complexes encircling only one duplex and able to diffuse much more rapidly than those forming a loop. The authors should discuss these observations and scenarios in more detail.

- The statement made on Page 6 on the value of the diffusion coefficient being "up to four orders of magnitude higher than the diffusion coefficients of many other DNA binding proteins" is not entirely correct. Single-molecule experiments on other clamps that topologically enclose DNA (e.g., PCNA) have resulted in diffusion coefficients that are very similar to the ones reported here.

- The authors remark on the fact that "Movements were not only seen in 750 mM NaCl but also in more physiological salt concentrations". Is there a dependence of the diffusion coefficient on salt concentration? Such dependence will inform on the mode of diffusion along the duplex (continuous versus discontinuous contact of protein and DNA).

- In the supplemental information, the authors describe the use of different oxygen scavenging strategies for different imaging experiments. The referencing to the different figure panels does not seem to make sense, though.

- Fig 2b: How many molecules are represented in the intensity histogram?
- Fig 2d: The authors should show distributions of the diffusion coefficient of a large number of molecules to allow the reader to assess the heterogeneity in diffusion rates. Also, the mean-square-displacement versus time graph suggests that the diffusion of that particular cohesin is bound (dashed line at $14 \mu\text{m}^2$). The authors should show whether this is the case for other molecules as well and discuss reasons for such bounded diffusion.
- Ext. Fig. 5: The authors' estimates of the size of the antibody-coated quantum dots are not realistic. The quantum dots themselves have a diameter of at most 10 nm (core, shell, and linker molecule), with the antibodies adding around 5 nm. Many excellent review papers discuss these geometries. Care should be taken in distinguishing between hydrodynamic radii and physical radii. It seems the authors used the former, while in this case the latter is more relevant. This is an important issue, since the true size of an antibody-coated quantum dot is not much larger than that of a nucleosome, invalidating their statement that while a quantum dot acts as a roadblock, a nucleosome likely doesn't.
- Figure 1f is insufficient as evidence for the cohesins not being able to pass each other. The data only shows one pair of fluorescent foci that did not exchange position during a limited time. The authors should use two differently labeled cohesins and provide more rigorous evidence.
- In the transcription experiments, the authors should introduce a ribonuclease and see whether the cohesins will move across a transcription complex. This simple experiment will show whether the transcription complex acts as a roadblock because of the high-molecular weight and randomly coiled RNA transcript. After all, given the fact that the size of the T7 RNA polymerase is only around 5 nm, it seems likely (based on the previous data) that the cohesin will pass a complex that consists of only the enzyme bound to the DNA.

Referee #3 (Remarks to the Author):

The cohesin ring topologically entraps DNA and regulates diverse chromosomal processes, including transcription, DNA repair, and chromosome segregation. Earlier studies in yeast have shown that the cohesin ring can be redistributed by active transcription along chromosomes, suggesting that, once topologically loaded, cohesin rings can slide on DNA. The dynamics and behavior of cohesin loaded on DNA have not been directly visualized, however. The current study by Davidson et al. fills this important gap. They used single-molecule TIRF microscopy to examine the dynamics of recombinant human cohesin spontaneously loaded onto lambda DNA in reconstituted systems in vitro. The major findings include: (1) Topologically loaded cohesin diffuses on DNA with surprisingly fast speed in an ATP-independent process; (2) Cohesin can slide past smaller obstacles (smaller than 10 nm), but not large ones (tens of nm), on DNA; (3) The insulator protein CTCF acts as an active barrier; and (4) Active transcription restricts the movement of cohesin on DNA to one direction. These findings provide further support for the topological embrace of DNA by cohesin and provide straightforward explanations for the well-documented mutual regulation between cohesin and the transcriptional machinery in vivo. For these reasons, this study presents a major conceptual advance in chromosome biology, and should be published in Nature.

I have only one major concern with the results presented in the study, which has to do with the role of ATP or the lack thereof. Although the authors provided strong evidence to suggest that human cohesin can topologically load onto DNA in the absence of the Scc2-Scc4 cohesin loader or ATP, this spontaneous entrapment of DNA by cohesin without ATP hydrolysis is surprising, and clearly differs from the in vivo situation. The authors stated in the paper that the recombinant fission yeast cohesin also underwent spontaneous loading in the absence of ATP. I could not find such data in the published work by Uhlmann and coworkers. In that study, ATP was included in the loading reactions.

To rule out the requirement of ATP hydrolysis in this spontaneous loading, the authors should test the loading and sliding behaviors of an ATPase-dead mutant of cohesin. Another important question is what happens when ATP or non-hydrolyzable ATP analogs are added to the bulk cohesin loading

and the single- molecule sliding assays. This needs to be tested experimentally. It is possible that ATP binding actually slows down the sliding of cohesin or renders it unable to slide past smaller obstacles, as the inner gate between Smc1 and Smc3 heads will be closed by ATP. This experiment is also important because the sliding assays involving transcription contained NTP (and thus ATP). One needs to separate the effects of ATP binding to cohesin from those of active transcription.

Minor points:

(1) On page 6, lines 4-5: I cannot find the photobleaching data on high-salt resistant cohesin bound to lambda DNA in extended Data Fig. 1f.

(2) An earlier study by Liu et al. (2015)(Mol. Cell 59, 426-436) showed that mitotic transcription restricts Sgo1 and possibly cohesin at inner centromeres, thereby strengthening centromeric cohesion. That finding can be nicely explained by transcription-dependent sliding of cohesin at mitotic centromeres. The authors should discuss this possibility

Authors' response

Response to referees' comments:

We thank all referees for carefully reviewing our manuscript and for their constructive criticism, which has helped us to improve our paper.

Referee #1

Specific comments:

1. *The facts that cohesin loading requires extremely low salt concentrations (50 mM), takes place in the absence of cohesin's SCC2/4 loading complex, and is independent of ATP raises strong doubts whether the DNA binding observed in vitro properly reconstitutes the loading of cohesin onto chromosomes in vivo. This is in contrast to previous in vitro DNA binding experiments with fission yeast cohesin, which showed a stimulation in binding by SCC2/4 and ATP (Murayama and Uhlmann, 2014, 2015). One possibility to support the notion that the authors observe a proper association of cohesin holocomplexes with DNA were to test whether, for example, isolated SMC1/3 dimers failed to load onto DNA in the same experimental setup.*

We agree with Referee #1 that the conditions required for cohesin to load onto DNA in our bulk and TIRF microscopy assays are different to those required in vivo. However, since it has so far been impossible to visualise single cohesin molecules binding to DNA in vivo, we feel that our in vitro system is useful to study such interactions, albeit in a simplified context. Furthermore, since the cohesin complexes we observe display the hallmarks of DNA entrapment, we maintain that these interactions are of relevance to those that occur in vivo.

Referee #1 contrasted our data on Scc2/4 and ATP independent cohesin loading with the in vitro findings of Murayama & Uhlmann, Nature 2014. While those authors do show Scc2/4 and ATP-dependent stimulation of salt-resistant binding of fission yeast cohesin to DNA, they also show that a small amount of salt resistant binding occurs in the absence of Scc2/4 and ATP and that this binding (at least in the absence of Scc2/4) is sensitive to DNA cleavage (Murayama & Uhlmann 2014, Fig. 3b). Although they did not test whether the DNA binding that occurs in the absence of ATP is sensitive to DNA or cohesin cleavage, the stimulation of cohesin:DNA binding by ATP addition is limited in the absence of Scc2/4 (Murayama & Uhlmann 2014, Extended Data Fig. 3d). This led the authors to speculate that fission yeast cohesin can spontaneously entrap DNA in vitro. This is consistent with our conclusions. We have performed a set of experiments to determine the requirements for ATP in our bulk and single molecule assays (see response to Referee #3, point 1). We find that ATP does not enhance cohesin loading in either assay, although an ATP binding deficient mutant form of cohesin shows a modest reduction in DNA binding in our bulk assay.

The referee correctly pointed out that our findings are in contrast to the situation *in vivo*, where ATP is required for cohesin loading. However, we do not think that this means that we are observing

artificial cohesin-DNA interaction in our *in vitro* assays, because these interactions possess all the characteristics of cohesin-DNA interactions *in vivo* (resistance to high salt, sensitivity to DNA and cohesin cleavage, requirement of all ring forming subunits to be present). Instead, we suspect that both ATP and the cohesin loading complex have catalytic roles in cohesin loading that increase the rate at which a reaction equilibrium is reached, but without affecting this equilibrium *per se*.

We thank Referee #1 for the suggestion to test whether Smc1/3 dimers associate with DNA in our TIRF microscopy setup. We have performed this experiment and found that Smc1/Smc3^{Halo-TMR} dimers do not bind to DNA when analysed using our highly sensitive custom built TIRF microscope (Extended Data Fig. 3d – f). This result provides further support for our conclusions that our assay measures physiologically relevant cohesin-DNA interactions which are presumably mediated by topological embracement of DNA by cohesin rings.

2. *Under the experimental conditions of the in vitro loading assay, most cohesin complexes tend to cluster non-specifically (with each other or with, for example, a bacteriophage RNA polymerase). It is conceivable that the observed behavior might be artificially biased by the clustering of these macromolecules. For example, might the inability of cohesin complexes to pass larger obstacles on the DNA be a consequence of their oligomerization, since one would, in principle, expect that a 50-nm cohesin ring should be able to traverse 20-nm protein barriers (Fig. 3c)? Only "some" cohesin molecules (pg. 6, Fig. 2) allowed real single-molecule analyses. If the authors could find conditions that increased the fraction of single molecule observations, then this could strongly support the conclusion that the assay measures biophysical properties of native cohesin complexes.*

Referee #1 commented that our manuscript would benefit from finding conditions in which single cohesin molecules could be more often observed, and was particularly concerned that the results of the roadblock assay might be affected by the oligomerisation state of cohesin.

We acknowledge that our data would be improved by using conditions in which only single cohesin molecules are analysed. To do this using a wide field of view and multiple channels is however technically challenging. To circumvent this we performed those experiments that required a wider field of view or multiple channels using a commercial TIRF microscopy setup and combined that with a basic characterization of the diffusive properties of single cohesin molecules using a custom built TIRF microscope. In response to the comments from Referee #1 we have extended our work by performing key “roadblock” experiments at high optical and temporal resolution using our custom built TIRF microscope. Using this setup, we again found that cohesin was able to bypass Halo-A488EcoRI^{E111Q} but not QDotEcoRI^{E111Q}. We believe that this provides additional support for our conclusion that this assay measures properties relevant to native cohesin complexes. Furthermore, we have used our custom built TIRF microscope in a number of new experiments (see response to Referee #1 point 1, Referee #1 point 3 and Referee #3 point 1).

3. *The manuscript would strongly benefit from more quantitative instead of qualitative analyses. One important control is that TEV cleavage of RAD21 releases cohesin from DNA. However, the only statement in the manuscript is that upon TEV cleavage, "less RAD21-Halo-TMR-TEV cohesin remained DNA-bound." (pg. 5 and Fig. 1f, g). To what degree was the amount of cohesin reduced in the case of the cleavable and non-cleavable RAD21 versions? A quantitative analysis can rule out the impression that also the levels of non-cleavable cohesin are strongly reduced in the presence of the protease (see the image shown in Fig. 1g). Similarly, it would be important to know (see above) which fraction of the cohesin signal corresponds to single cohesin complexes (pg. 6). Comparison of the fluorescence intensities of single cohesin molecules and cohesin clusters might furthermore allow an estimate of the stoichiometry of cohesin in the clusters.*

Referee #1 commented that our manuscript would benefit from more quantitative analyses and was concerned that TEV protease had a deleterious effect on non-cleavable cohesin (initial manuscript Fig. 1 g). We have performed a new experiment in which we incubated a portion of bead-bound cohesin with TEV protease during purification and then washed away the protease prior to elution. We then compared this cleaved cohesin with non-cleaved cohesin, purified in parallel, using our custom built TIRF microscope. Cleaved cohesin bound to DNA much less efficiently than intact cohesin (see Extended Data Fig. 3a – c). This analysis was performed in low salt binding buffer.

Under these conditions non-cleaved cohesin is present on DNA as multimeric complexes, making it difficult to accurately determine the absolute number of molecules by monitoring the photobleaching step size. As an alternative, we quantified the number of these cohesin structures per DNA, and found a dramatic difference between the two forms of cohesin (see Extended Data Fig. 3 c). We believe that this quantitative data coupled with the more qualitative data shown in Fig. 1f strongly supports the idea that TEV cleavage releases cohesin from DNA in our assay.

4. *In the initial T7 experiments, cohesin clusters seem to translocate with the polymerases (Fig. 4d). This supports the notion that transcription might 'push' (or 'pull') cohesin. However, these experiments have been performed solely under low salt conditions, which suggests that the majority of cohesin complexes observed do not encircle DNA and might rather bind directly (non-specifically) to the RNA polymerases. For the experiments that follow, the authors perform a high salt wash and hence select for cohesin rings that have encircled DNA. Under these conditions, however, only a small fraction of T7 polymerases seem to actively transcribe DNA ("Occasionally, cohesin-Halo-TMR T7 RNAP complexes converted to unidirectional movement", pg. 9 and Fig. 4e). Without further evidence that transcription takes place under these conditions, the conclusion that transcription moves cohesin cannot be made. Is this movement dependent on NTPs? One possible control were to inhibit T7 polymerase (for example by T7 lysozyme) and test whether this prevented unidirectional movement of cohesin-T7 polymerase complexes. Since the authors observe an accumulation of cohesin at T7 terminator sequences in vivo (Extended Data Fig. 6c), the authors could also insert the same sequence at a defined position within the DNA template and test whether this results in the accumulation of cohesin at this position in vitro.*

The referee questioned whether T7 RNAP was able to transcribe in the conditions described in Figure 4e, i.e. after a high salt wash to select for entrapping cohesin. We have modified the text to clarify the point raised here. In Figure 4e, the high salt wash to select for entrapping cohesin complexes was followed by a wash with the same T7 transcription buffer used in Figure 4a and d. Since the T7 RNAP movement observed in Fig. 4a was dependent on NTPs, we have no reason to suspect that the movement observed in Fig. 4e is NTP independent. Unfortunately due to time constraints we were unable to perform the additional experiments suggested here. However we would like to point out that there is really no reason for doubting that the unidirectional, NTP dependent movement of T7 RNAP we describe is caused by transcription.

5. *The author suggest that CTCF acts as a specific roadblock for cohesin movement (Fig. 4 g, h). However, it is likely that any protein that would form a large DNA loop would create an obstacle for the passage of a protein that slides along DNA. Since the assay cannot address whether cohesin can support loop formation at CTCF sites, for example by encircling DNA sequences in cis, it is not obvious how these experiments add to explaining how a chromatin domain might be formed by CTCF and cohesin in vivo. One possibility to test whether CTCF can specifically prevent the translocation of cohesin would be to either replace the 4 CTCF binding sites by a single site, assuming that this would not result in loop formation, or to use a mutant version of CTCF that cannot multimerize (Bonchuk et al., 2015).*

Referee #1 questions the usefulness of our cohesin/CTCF assay in understanding how these proteins function together to form chromatin domains in vivo, principally since we cannot determine whether cohesin and CTCF can form loops in our assay. The referee suggests performing an experiment using a DNA template with one CTCF binding site instead of four. We have modified our text to more clearly state our interpretation of this data. We do not use this assay as a test for any DNA looping properties of cohesin/CTCF, but rather as a system to study how cohesin might be positioned by CTCF in vivo. Our data is consistent with the hypothesis that cohesin translocates relative to DNA until it encounters CTCF, which acts as a "roadblock" to cohesin translocation.

We agree that the interpretation of these experiments is complicated by the fact that we used a DNA template that contains four closely spaced CTCF binding sites. We therefore have repeated these experiments using a template that contains a single CTCF binding site and found that CTCF still acts as a significant barrier to cohesin translocation (Extended Data Fig. 10f). We would very much like to extend our study into the mechanism of CTCF/cohesin dependent loop / chromatin domain

formation, however this is out with the scope of this manuscript

6. *The authors claim that the cohesin complexes they use are ring-shaped (pg. 4). Even though they show that the four cohesin subunits assemble into a complex (Fig. 1a), the authors didn't directly test for ring formation. This could, for example, be achieved by testing whether the two fragments of RAD21 generated by TEV cleavage remain associated with an SMC1/3 dimer.*

The referee comments that we do not know whether our recombinant cohesin complexes are ring-shaped, and suggests testing this by determining if the Rad21 cleavage fragments produced after TEV protease cleavage remain associated with Smc1/3. We have clarified this point in the text. The data showing that the Rad21^{Halo} cleavage fragments remain associated with Smc1/3 can be found in Fig. 1c and again in the newly added Extended Data Fig. 3a. In both experiments, Rad21^{Halo} cohesin was immobilized on anti-Flag agarose via Smc3^{Flag}, incubated with TEV protease, washed extensively and then eluted with Flag peptide. The presence of Rad21^{Halo} cleavage fragments (visualised by silver staining, western blotting and TMR excitation) in the eluate demonstrates that they remain associated with Smc1/3 and therefore that our non-cleaved cohesin complexes must have formed rings. Rad21^{GFP} cohesin expressed and purified using identical methods has been characterized in this way and also via rotary shadowing electron microscopy in Huis in 't Veld et al 2014.

7. *To allow readers to judge the efficiency of the co-purification of a "small amount" (pg. 4) of circular DNA with cohesin, the authors should plot the 'fractions of input' in Fig. 1b and d instead of the 'fold change over no cohesin control'.*

The referee suggests that we modify Fig 1b and 1d to plot fractions of input DNA rather than fold change over no cohesin control. We agree that fold-change is confusing for readers and have modified the figures accordingly. We acknowledge that cohesin binds a very small percentage of total DNA in this assay (particularly in Fig. 1b and 1d, less so in the newly added experiments shown in Extended Data Fig. 2c and 2d). We have provided new data that demonstrates that we can significantly improve the recovery of DNA by treating the cohesin:DNA antibody beads with proteinase K directly rather than by eluting with antigenic peptide (Extended Data Fig. 1a). We are unfortunately unable to perform this elution method in all experiments since it consumes a prohibitive amount of antibody beads.

8. *Taking into consideration that the effect of T7 RNA polymerase transcription on the positioning of cohesin on yeast chromosomes has already been described in the supporting manuscript (Ocampo-Hafalla et al.), reproduction of the data shown in Fig. 8b of this manuscript in Extended Fig. 6a and b is redundant.*

Referee #1 asks that we remove Extended Data Figure 6 since it is reproduced in Ocampo-Hafalla et al 2016. We agree that this data is indeed similar to that described in Ocampo-Hafalla et al 2016. However this exact experiment, which demonstrates that cohesin accumulates at a T7 termination site downstream of a gene under the control of a T7 promoter in a strain that expresses T7 RNAP, is not reproduced in that manuscript. Since this in vivo finding fits excellently with our data, we would very much like to include it in our current manuscript.

9. *The authors refer to the SCC2/4 cohesin loading complex as CLC. Since the same complex has previously been named 'adherin' or 'kollerin' by different authors, introducing yet another name is very confusing.*

The referee asks that we do not refer to the cohesin loading complex as CLC but instead to one of its preexisting names. We agree that it would be confusing to introduce yet another name for the cohesin loading complex, but this was not our intention. We simply used "CLC" as an abbreviation for cohesin loading complex (a term widely used in the literature) in an attempt to reduce the word count of our manuscript. However, we agree that the use of this abbreviation makes the text more difficult to read and have therefore changed the abbreviation back to "cohesin loading complex" in all cases.

Referee #2

1. *The authors show that cohesin compacts singly-tethered DNA and leaves doubly-tethered experiments in a stretched state. For these experiments to be interpreted correctly in the context of incompletely stretched DNA molecules supporting looping by cohesin, the authors should provide information on the actual tether length of the DNA molecules analyzed. This is particularly important when analyzing the behavior of cohesins that seem to be loaded on doubly-tethered DNA: they may correspond to cohesins encircling a single copy of the duplex, or they may encircle two duplexes, holding together a loop. The latter scenario would be consistent with the authors' statement on page 6 that some spots contain multiple copies of cohesin. The simultaneous observation of cohesin signal and a colocalized spot of DNA stain may demonstrate the existence of such loops. Relatedly, the complexes with the highest diffusion coefficient should correspond with spots that contain only one cohesin tetramer; these would be the complexes encircling only one duplex and able to diffuse much more rapidly than those forming a loop. The authors should discuss these observations and scenarios in more detail.*

Referee #2 asks us to provide information as to the actual tether length of doubly tethered DNA molecules, to comment on the possibility that DNA loops might form and that cohesin might entrap more than one DNA duplex. We would like to thank the referee for pointing out these interesting possibilities, which we are indeed planning to study in the future. As suggested by the Referee, we have now measured contour lengths of doubly tethered DNA molecules, found that these are largely but not fully extended (to ~ 67%, reported on page 4 of the revised manuscript; please see Figure 1 below), indicating that such loops could indeed be formed. However, under our imaging conditions, we could not obtain any evidence for their formation. We therefore feel that a thorough analysis of the ability of cohesin and/or CTCF to generate DNA loop formation *in vitro* is beyond the scope of our current manuscript, in which our main aim was to characterize cohesin translocation along DNA, a phenomenon which we can clearly observe for both single and multimeric cohesin structures, and the properties of which we have characterized in detail in this study.

Figure 1. Tether length of doubly tethered λ -phage DNA. Median and interquartile range are shown; $n = 54$.

2. *The statement made on Page 6 on the value of the diffusion coefficient being "up to four orders of magnitude higher than the diffusion coefficients of many other DNA binding proteins" is not entirely correct. Single-molecule experiments on other clamps that topologically enclose DNA (e.g., PCNA) have resulted in diffusion coefficients that are very similar to the ones reported here.*

The referee comments that the diffusion coefficient calculated for other sliding clamps is similar to the one we report for cohesin. We thank the referee for highlighting this, and have added this information to the text.

3. *The authors remark on the fact that "Movements were not only seen in 750 mM NaCl but also in more physiological salt concentrations". Is there a dependence of the diffusion coefficient on salt concentration? Such dependence will inform on the mode of diffusion along the duplex (continuous versus discontinuous contact of protein and DNA).*

The referee asks whether there is a dependence of cohesin's diffusion coefficient on salt concentration. We have now described in the figure legend of Extended Data Fig. 1 that little diffusion is observed upon incubation of cohesin with DNA at a salt concentration of 50 mM, suggesting that the diffusion coefficient of cohesin on DNA is indeed dependent on salt concentration. We attempted to gather a set of data that will allow us to determine the diffusion coefficient of cohesin at 150 mM salt; unfortunately this is time consuming and we are unable to include the data at this point.

4. *In the supplemental information, the authors describe the use of different oxygen scavenging strategies for different imaging experiments. The referencing to the different figure panels does not seem to make sense, though.*

Referee #2 noted that the references to the figure panels in which oxygen scavenger systems were used are incorrect. We thank the referee for pointing out this inaccuracy which we have corrected.

5. *Fig 2b: How many molecules are represented in the intensity histogram?*

The referee asks how many molecules were analysed in the histogram in Figure 2b. We have added this information to the figure legend of Figure 2b. Data was collected from 290 regions that were identified as having a fluorescent signal and 228 background regions that did not contain fluorescence.

6. *Fig 2d: The authors should show distributions of the diffusion coefficient of a large number of molecules to allow the reader to assess the heterogeneity in diffusion rates. Also, the mean-square-displacement versus time graph suggests that the diffusion of that particular cohesin is bound (dashed line at 14 micron²). The authors should show whether this is the case for other molecules as well and discuss reasons for such bounded diffusion.*

The referee asks that we show the MSD of a number of different cohesin molecules and asks why the graph originally shown in Figure 2d reaches a plateau at $\sim 14 \mu\text{m}^2$. We agree with the referee that this is an important point and one that was not adequately described in the manuscript. We have replaced Fig 2d with a new panel in which the diffusion coefficients of cohesin \pm ATP and an ATP binding and hydrolysis deficient form of cohesin are compared. Figure 2d represents the combined data on the diffusion of individual cohesin molecules (cohesin wt – ATP: 13 molecules, cohesin wt + ATP: 18 molecules, cohesin KA mutant + ATP: 28 molecules). We found that MSD plots of individual cohesin complexes show strong fluctuations and cannot be well approximated by normal or bound diffusion. This indicates that at the level of individual complexes statistical fluctuations dominate the random movement and do not allow the reliable extraction of diffusion coefficients of single diffusing molecules. Therefore, we combined data from individual molecules and calculated the average MSD value for each time-point (Extended Data Fig. 2f). Unlike the individual plots, the combined plot shows behaviour typical for bound diffusion and allows for the extraction of the diffusion coefficient during normal diffusion by determining the slope of the linear part of the graph (data plotted in Fig. 2d). Therefore, in the revised manuscript we decided to use only data combined from single cohesin molecules in order to calculate diffusion coefficients.

As also pointed out by the referee, we indeed observe that the MSD plot plateaus. Bound diffusion is typical in systems where diffusion is confined to a certain area. In our case, cohesin molecules undergo one-dimensional diffusion along doubly tethered DNA molecules and the ends of the DNA immobilized on the coverslip naturally restrict their movements. We also observed that other cohesin complexes present on DNA, which decreased effective area allowed for the diffusion further, could also restrict the diffusion of cohesin molecules. We have added discussion of this point to the figure legend of Extended Data Fig. 2f.

7. *Ext. Fig. 5: The authors' estimates of the size of the antibody-coated quantum dots are not realistic. The quantum dots themselves have a diameter of at most 10 nm (core, shell, and*

linker molecule), with the antibodies adding around 5 nm. Many excellent review papers discuss these geometries. Care should be taken in distinguishing between hydrodynamic radii and physical radii. It seems the authors used the former, while in this case the latter is more relevant. This is an important issue, since the true size of an antibody-coated quantum dot is not much larger than that of a nucleosome, invalidating their statement that while a quantum dot acts as a roadblock, a nucleosome likely doesn't.

Referee #2 comments that our estimate as to the diameter of the QDot antibody conjugates used in Figure 3 is inaccurate, and questions our assertion that nucleosomes are unlikely to act as a barrier to cohesin movement. We thank the referee for highlighting this inaccuracy; we have modified the text accordingly. In addition, we have performed new roadblock experiments using recombinant nucleosomes. These data show that, unlike $^{QDot}EcoRI^{E111Q}$, a nucleosome does not act as a non-diffusible barrier to cohesin (Fig. 3c).

8. *Figure 1f is insufficient as evidence for the cohesins not being able to pass each other. The data only shows one pair of fluorescent foci that did not exchange position during a limited time. The authors should use two differently labeled cohesins and provide more rigorous evidence.*

The referee states that the evidence provided in Extended Data Fig. 1f is insufficient to claim that cohesin molecules do not pass one another, and suggests an experiment in which two differently coloured forms of cohesin are used to test this idea more rigorously. We acknowledge that the data provided is insufficient to claim that two cohesin molecules never pass one another on DNA. Unfortunately due to the technically challenging nature of dual colour single molecule imaging we were unable to generate sufficient data to include it in this version of the manuscript and so have removed this statement from the text

9. *In the transcription experiments, the authors should introduce a ribonuclease and see whether the cohesins will move across a transcription complex. This simple experiment will show whether the transcription complex acts as a roadblock because of the high-molecular weight and randomly coiled RNA transcript. After all, given the fact that the size of the T7 RNA polymerase is only around 5 nm, it seems likely (based on the previous data) that the cohesin will pass a complex that consists of only the enzyme bound to the DNA.*

Referee #2 suggests that we use RNase to test whether cohesin translocation is affected by T7 RNAP directly or the presence of an RNA transcript. We thank the referee for suggesting this very interesting experiment. We agree that it is likely that the RNA transcript rather than T7 RNAP *per se* acts as a roadblock to cohesin translocation and have added a sentence to the text to highlight this possibility. Unfortunately, we were unable to perform this experiment due to time constraints.

Referee #3

1. *I have only one major concern with the results presented in the study, which has to do with the role of ATP or the lack thereof. Although the authors provided strong evidence to suggest that human cohesin can topologically load onto DNA in the absence of the Scc2-Scc4 cohesin loader or ATP, this spontaneous entrapment of DNA by cohesin without ATP hydrolysis is surprising, and clearly differs from the *in vivo* situation. The authors stated in the paper that the recombinant fission yeast cohesin also underwent spontaneous loading in the absence of ATP. I could not find such data in the published work by Uhlmann and coworkers. In that study, ATP was included in the loading reactions.*

To rule out the requirement of ATP hydrolysis in this spontaneous loading, the authors should test the loading and sliding behaviors of an ATPase-dead mutant of cohesin. Another important question is what happens when ATP or non-hydrolyzable ATP analogs are added to the bulk cohesin loading and the single-molecule sliding assays. This needs to be tested experimentally. It is possible that ATP binding actually slows down the sliding of cohesin or renders it unable to slide past smaller obstacles, as the inner gate between Smc1 and Smc3 heads will be closed by ATP. This experiment is also important because the sliding assays involving transcription contained NTP (and thus ATP). One needs to separate the effects of ATP binding to cohesin from those of active transcription.

Referee #3 comments that we do not accurately report the findings of Murayama and Uhlmann (Murayama & Uhlmann, Nature 2014) and asks us to determine the ATP requirements of cohesin loading in our bulk and TIRF assays. We thank Referee #3 for highlighting this issue. We have modified the text to more clearly state the findings of Murayama and Uhlmann (namely that DNA cleavage-sensitive DNA binding of fission yeast cohesin was not tested in the absence of ATP but was demonstrated in the absence of Scc2/4). Furthermore, we have performed a set of experiments to determine the role of ATP on the salt resistant binding of cohesin to DNA in our bulk and TIRF microscopy assays. We find that the presence of ATP is not required for salt-resistant cohesin:DNA interactions (Extended Data Fig. 2c, d) and does not measurably affect the diffusion coefficient of single cohesin complexes (Fig 2d, Extended Data Fig. 2e, f). We have also tested the DNA binding of an ATP binding/hydrolysis deficient form of cohesin (Smc1/3 KA). KA cohesin reproducibly bound less DNA than wildtype cohesin in our bulk assay, suggesting that ATP binding/hydrolysis might enhance but is not required for cohesin:DNA interactions in vitro. KA cohesin also bound to DNA in our TIRF microscopy assay and the diffusion coefficient of single molecules of KA cohesin was indistinguishable from wildtype cohesin.

2. *On page 6, lines 4-5: I cannot find the photobleaching data on high-salt resistant cohesin bound to lambda DNA in Extended Data Fig. 1f.*

Referee #3 comments that the reference to photobleaching data in Extended Data Figure 1f is confusing. We have modified the text accordingly.

3. *An earlier study by Liu et al. (2015) (Mol. Cell 59, 426-436) showed that mitotic transcription restricts Sgo1 and possibly cohesin at inner centromeres, thereby strengthening centromeric cohesion. That finding can be nicely explained by transcription-dependent sliding of cohesin at mitotic centromeres. The authors should discuss this possibility.*

The referee suggests that the mitotic transcription-dependent restriction of Sgo1 and possibly cohesin at the inner centromere can be explained by transcription-dependent sliding, and suggests that we discuss this possibility in the text. We thank Referee #3 for this interesting suggestion. We have added a sentence to the discussion section and cited the relevant paper.

Previous Referee Reports – round 2

Referee #1 (Remarks to the Author):

In their revised manuscript, Peters and colleagues have addressed the majority of the concerns raised during the previous round of reviews. The new experiments support the conclusion that the behavior of the purified cohesin complexes in vitro recapitulates their in vivo characteristics. In addition to the control experiments presented already in the original manuscript, the authors now demonstrate that (1) association with DNA in the flow cells requires cohesin holocomplexes, (2) cohesin can slide past nucleosomes, and (3) a single CTCF site can block cohesin passage, ruling out that merely CTCF-induced higher order DNA conformations obstruct cohesin movement. Even though in vitro cohesin loading still requires very low salt conditions and is, as the authors now demonstrate further, largely ATP- independent, the additional control experiments greatly help to strengthen the biological relevance of the authors' findings.

In the meantime, Koshland, Greene and colleagues have described sliding of fission yeast cohesin complexes along DNA using a similar, yet not identical, single- molecule imaging setup (Stigler et al., 2016). The work described in the current manuscript using human cohesin complexes, does, however, go beyond the published work. First, the experiments presented include additional important controls (e.g. cohesin ring opening) and use a labelling method that is much less likely to influence the properties of diffusing cohesin complexes (a small molecule fluorophore instead of QDs). Second, Peters and colleagues show that an RNA polymerase, and not just a random DNA motor protein such as FtsK, translocates cohesin. This result greatly strengthens the hypothesis that transcription can move cohesin along a chromosome. Third, the finding that CTCF can act as a

roadblock for cohesin movement is a novel aspect, which has not been addressed in the previous publication (due to the absence of a CTCF homolog in yeast).

Nevertheless, the main conclusions of the two papers, namely that cohesin is able to diffuse along DNA helices and past small but not large obstacles, are identical. The rates of movement of yeast and human cohesin are remarkably similar under comparable assay conditions (a few $\mu\text{m}^2/\text{s}$) and both complexes can pass obstacles of 10 nm but not of 20 nm. While the work warrants, without doubt, publication, much of the novelty of both, the experimental approach and the results obtained, is no longer obvious to the broad audience of Nature. I therefore recommend publication of the work in a more specialized journal after the authors have addressed two remaining issues:

1. Compared to the work from the Koshland and Greene labs, the most important new aspect described by Peters and colleagues is the movement of cohesin with T7 RNA polymerase. What puzzles me is that cohesin and T7 RNA pol always co-localize, even when one component is added later than the other or even in the

absence of NTPs, when T7 RNA pol should not be active (Fig. 4d and e). If T7 RNA pol indeed pushed or pulled cohesin, one would expect to see that both components should only move together once the polymerase 'bumps' into cohesin, similar to what has been observed with FtsK. The authors argue that they didn't have sufficient time to test the dependency of the movement of cohesin-RNA pol complexes on active transcription. Taking into account the importance of this conclusion, I would like to strongly encourage the authors to again consider a control experiment in which transcription is selectively prevented (for example by adding a T7 RNA pol inhibitor or using a catalytically inactive T7 RNA pol mutant).

2. Since Koshland, Greene and colleagues found that yeast cohesin 'pauses' at nucleosomal DNA sites and from this concluded that nucleosomes act as 'semipermeable' barriers for cohesin movement, I wondered whether the authors observed in the high temporal resolution data whether human cohesin behaves in a similar manner.

Referee #2 (Remarks to the Author): The impact of this manuscript in terms of novelty has been significantly decreased by the recent publication of single-molecule studies on cohesin by the Greene group in Cell Reports a few months ago. Further, it has been frustrating to read through the rebuttal written by the authors of the current manuscript: all my suggestions for additional experiments are welcomed as important and relevant, but none of them have been performed (too difficult, outside the scope of the study, time constraints). I don't think the current manuscript provides the molecular mechanistic insight that would warrant publication in Nature. The conclusions that are drawn are not justified based on the data shown (in line with my previous comments).

Authors' response

Response to referees' comments:

Referee #1

1. *Compared to the work from the Koshland and Greene labs, the most important new aspect described by Peters and colleagues is the movement of cohesin with T7 RNA polymerase. What puzzles me is that cohesin and T7 RNA pol always co-localize, even when one component is added later than the other or even in the absence of NTPs, when T7 RNA pol should not be active (Fig. 4d and e). If T7 RNA pol indeed pushed or pulled cohesin, one would expect to see that both components should only move together once the polymerase 'bumps' into cohesin, similar to what has been observed with FtsK. The authors argue that they didn't have sufficient time to test the dependency of the movement of cohesin-RNA pol complexes on active transcription. Taking into account the importance of this conclusion, I would like to strongly encourage the authors to again consider a control experiment in which transcription is selectively prevented (for example by adding a T7 RNA pol inhibitor or using a catalytically inactive T7 RNA pol mutant).*

We thank Referee #1 for highlighting that cohesin and T7RNAP colocalise and for the suggestion to determine the effect of T7RNAP on cohesin translocation in conditions in which transcription is inhibited. Unfortunately the low ionic strength in which T7RNAP is transcriptionally active and in which cohesin binds to DNA likely contributes to the T7RNAP and cohesin colocalisation observed in our assay. This colocalisation means that we cannot currently determine whether T7RNAP pushes or pulls cohesin, but can address whether T7RNAP affects cohesin translocation *per se*. Since unidirectional T7RNAP movement was only detectable in the presence of NTPs, we argue that this most likely represents transcribing T7RNAP complexes. Unidirectional movement of T7RNAP-cohesin complexes was also detectable in the presence of NTPs, a form of translocation never observed for cohesin alone, suggesting that T7RNAP transcription occurs here and that this affects the translocation of cohesin complexes. We think that, under these assay conditions, experiments in which the transcriptional activity of T7RNAP is inhibited would not provide additional insight into how T7RNAP constrains cohesin translocation.

2. *Since Koshland, Greene and colleagues found that yeast cohesin 'pauses' at nucleosomal DNA sites and from this concluded that nucleosomes act as 'semipermeable' barriers for cohesin movement, I wondered whether the authors observed in the high temporal resolution data whether human cohesin behaves in a similar manner.*

Referee #1 points out that Stigler et al find that fission yeast cohesin 'pauses' at nucleosomes and asks whether we observe a similar phenomenon when human cohesin encounters nucleosomes. We have not visualised the passage of cohesin past nucleosomes at high temporal resolution so cannot provide data as to whether cohesin 'pauses' at these sites. However, we do observe such a phenomenon when cohesin encounters Halo-A488-EcoRI^{E111Q} (see Extended Data Fig. 8a for examples), as also reported for fission yeast cohesin (Stiegler et al 2016). This suggests that, like fission yeast cohesin, human cohesin likely pauses at nucleosomes.

Authors' response

Referee #1

1. We agree with Referee #1 that the conditions required for cohesin to load onto DNA in our bulk and TIRF microscopy assays are different to those required *in vivo*. However, since it has so far been impossible to visualise single cohesin molecules binding to DNA *in vivo*, we feel that our *in vitro* system is useful to study such interactions, albeit in a simplified context. Furthermore, since the cohesin complexes we observe display the hallmarks of DNA entrapment, we maintain that these interactions are of relevance to those that occur *in vivo*.

Referee #1 contrasted our data on Scc2/4 and ATP independent cohesin loading with the *in vitro* findings of Murayama & Uhlmann, Nature 2014. While those authors do show Scc2/4 and ATP-dependent stimulation of salt-resistant binding of fission yeast cohesin to DNA, they also show that a small amount of salt resistant binding occurs in the absence of Scc2/4 and ATP and that this binding (at least in the absence of Scc2/4) is sensitive to DNA cleavage (Murayama & Uhlmann 2014, Fig. 3b). Although they did not test whether the DNA binding that occurs in the absence of ATP is sensitive to DNA or cohesin cleavage, the stimulation of cohesin:DNA binding by ATP addition is limited in the absence of Scc2/4 (Murayama & Uhlmann 2014, Extended Data Fig. 3d). This led the authors to speculate that fission yeast cohesin can spontaneously entrap DNA *in vitro*. This is consistent with our conclusions. We have performed a set of experiments to determine the requirements for ATP in our bulk and single molecule assays (see response to Referee #3, point 1). We find that ATP does not enhance cohesin loading in either assay, although an ATP binding deficient mutant form of cohesin shows a modest reduction in DNA binding in our bulk assay.

The referee correctly pointed out that our findings are in contrast to the situation *in vivo*, where ATP is required for cohesin loading. However, we do not think that this means that we are observing artificial cohesin-DNA interaction in our *in vitro* assays, because these interactions possess all the characteristics of cohesin-DNA interactions

in vivo (resistance to high salt, sensitivity to DNA and cohesin cleavage, requirement of all ring forming subunits to be present). Instead, we suspect that both ATP and the cohesin loading complex have catalytic roles in cohesin loading that increase the rate at which a reaction equilibrium is reached, but without affecting this equilibrium *per se*.

We thank Referee #1 for the suggestion to test whether Smc1/3 dimers associate with DNA in our TIRF microscopy setup. We have performed this experiment and found that Smc1/Smc3Halo-TMR dimers do not bind to DNA when analysed using our highly sensitive custom built TIRF microscope (Extended Data Fig. 3d – f). This result provides further support for our conclusions that our assay measures physiologically relevant cohesin-DNA interactions which are presumably mediated by topological embracement of DNA by cohesin rings.

2. Referee #1 commented that our manuscript would benefit from finding conditions in which single cohesin molecules could be more often observed, and was particularly concerned that the results of the roadblock assay might be affected by the oligomerisation state of cohesin.

We acknowledge that our data would be improved by using conditions in which only single cohesin molecules are analysed. To do this using a wide field of view and multiple channels is however technically challenging. To circumvent this we performed those experiments that required a wider field of view or multiple channels using a commercial TIRF microscopy setup and combined that with a basic characterization of the diffusive properties of single cohesin molecules using a custom built TIRF microscope. In response to the comments from Referee #1 we have extended our work by performing key “roadblock” experiments at high optical and temporal resolution using our custom built TIRF microscope. Using this setup, we again found that cohesin was able to bypass Halo-A488EcoRIE111Q but not QDotEcoRIE111Q. We believe that this provides additional support for our conclusion that this assay measures properties relevant to native cohesin complexes. Furthermore, we have used our custom built TIRF microscope in a number of new experiments (see response to Referee #1 point 1, Referee #1 point 3 and Referee #3 point 1).

3. Referee #1 commented that our manuscript would benefit from more quantitative analyses and was concerned that TEV protease had a deleterious effect on noncleavable cohesin (initial manuscript Fig. 1 g).

We have performed a new experiment in which we incubated a portion of bead-bound cohesin with TEV protease during purification and then washed away the protease prior to elution. We then compared this cleaved cohesin with non-cleaved cohesin, purified in parallel, using our custom built TIRF microscope. Cleaved cohesin bound to DNA much less efficiently than intact cohesin (see Extended Data Fig. 3a – c). This analysis was performed in low salt binding buffer. Under these conditions non-cleaved cohesin is present on DNA as multimeric complexes, making it difficult to accurately determine the absolute number of molecules by monitoring the photobleaching step size. As an alternative, we quantified the number of these cohesin structures per DNA, and found a dramatic difference between the two forms of cohesin (see Extended Data Fig. 3 c). We believe that this quantitative data coupled with the more qualitative data shown in Fig. 1f strongly supports the idea that TEV cleavage releases cohesin from DNA in our assay.

4. The referee questioned whether T7 RNAP was able to transcribe in the conditions described in Figure 4e, i.e. after a high salt wash to select for entrapping cohesin.

We have modified the text to clarify the point raised here. In Figure 4e, the high salt wash to select for entrapping cohesin complexes was followed by a wash with the same T7 transcription buffer used in Figure 4a and d. Since the T7 RNAP movement observed in Fig. 4a was dependent on NTPs, we have no reason to suspect that the movement observed in Fig. 4e is NTP independent. Unfortunately due to time

constraints we were unable to perform the additional experiments suggested here. However we would like to point out that there is really no reason for doubting that the unidirectional, NTP dependent movement of T7 RNAP we describe is caused by transcription.

5. Referee #1 questions the usefulness of our cohesin/CTCF assay in understanding how these proteins function together to form chromatin domains *in vivo*, principally since we cannot determine whether cohesin and CTCF can form loops in our assay. The referee suggests performing an experiment using a DNA template with one CTCF binding site instead of four.

We have modified our text to more clearly state our interpretation of this data. We do not use this assay as a test for any DNA looping properties of cohesin/CTCF, but rather as a system to study how cohesin might be positioned by CTCF *in vivo*. Our data is consistent with the hypothesis that cohesin translocates relative to DNA until it encounters CTCF, which acts as a “roadblock” to cohesin translocation.

We agree that the interpretation of these experiments is complicated by the fact that we used a DNA template that contains four closely spaced CTCF binding sites. We therefore have repeated these experiments using a template that contains a single CTCF binding site and found that CTCF still acts as a significant barrier to cohesin translocation (Extended Data Fig. 10f). We would very much like to extend our study into the mechanism of CTCF/cohesin dependent loop / chromatin domain formation, however this is out with the scope of this manuscript

6. The referee comments that we do not know whether our recombinant cohesin complexes are ring-shaped, and suggests testing this by determining if the Rad21 cleavage fragments produced after TEV protease cleavage remain associated with Smc1/3.

We have clarified this point in the text. The data showing that the Rad21Halo cleavage fragments remain associated with Smc1/3 can be found in Fig. 1c and again in the newly added Extended Data Fig. 3a. In both experiments, Rad21Halo cohesin was immobilized on anti-Flag agarose via Smc3Flag, incubated with TEV protease, washed extensively and then eluted with Flag peptide. The presence of Rad21Halo cleavage fragments (visualised by silver staining, western blotting and TMR excitation) in the eluate demonstrates that they remain associated with Smc1/3 and therefore that our non-cleaved cohesin complexes must have formed rings. Rad21GFP cohesin expressed and purified using identical methods has been characterized in this way and also via rotary shadowing electron microscopy in Huis in ‘t Veld et al 2014.

7. The referee suggests that we modify Fig 1b and 1d to plot fractions of input DNA rather than fold change over no cohesin control.

We agree that fold-change is confusing for readers and have modified the figures accordingly. We acknowledge that cohesin binds a very small percentage of total DNA in this assay (particularly in Fig. 1b and 1d, less so in the newly added experiments shown in Extended Data Fig. 2c and 2d). We have provided new data that demonstrates that we can significantly improve the recovery of DNA by treating the cohesin:DNA antibody beads with proteinase K directly rather than by eluting with antigenic peptide (Extended Data Fig. 1a). We are unfortunately unable to perform this elution method in all experiments since it consumes a prohibitive amount of antibody beads.

8. Referee #1 asks that we remove Extended Data Figure 6 since it is reproduced in Ocampo-Hafalla et al 2016.

We agree that this data is indeed similar to that described in Ocampo-Hafalla et al 2016. However this exact experiment, which demonstrates that cohesin accumulates at a T7 termination site downstream of a gene under the control of a T7 promoter in a

strain that expresses T7 RNAP, is not reproduced in that manuscript. Since this *in vivo* finding fits excellently with our data, we would very much like to include it in our current manuscript.

9. The referee asks that we do not refer to the cohesin loading complex as CLC but instead to one of its preexisting names.

We agree that it would be confusing to introduce yet another name for the cohesin loading complex, but this was not our intention. We simply used “CLC” as an abbreviation for cohesin loading complex (a term widely used in the literature) in an attempt to reduce the word count of our manuscript. However, we agree that the use of this abbreviation makes the text more difficult to read and have therefore changed the abbreviation back to “cohesin loading complex” in all cases.

Referee #2

1. Referee #2 asks us to provide information as to the actual tether length of doubly tethered DNA molecules, to comment on the possibility that DNA loops might form and that cohesin might entrap more than one DNA duplex.

We would like to thank the referee for pointing out these interesting possibilities, which we are indeed planning to study in the future. As suggested by the Referee, we have now measured contour lengths of doubly tethered DNA molecules, found that these are largely but not fully extended (to ~ 67%, reported on page 4 of the revised manuscript; please see Figure 1 below), indicating that such loops could indeed be formed. However, under our imaging conditions, we could not obtain any evidence for their formation. We therefore feel that a thorough analysis of the ability of cohesin and/or CTCF to generate DNA loop formation *in vitro* is beyond the scope of our current manuscript, in which our main aim was to characterize cohesin translocation along DNA, a phenomenon which we can clearly observe for both single and multimeric cohesin structures, and the properties of which we have characterized in detail in this study.

Figure 1. Tether length of doubly tethered λ -phage DNA. Median and interquartile range are shown; $n = 54$.

2. The referee comments that the diffusion coefficient calculated for other sliding clamps is similar to the one we report for cohesin.

We thank the referee for highlighting this, and have added this information to the text.

3. The referee asks whether there is a dependence of cohesin's diffusion coefficient on salt concentration.

We have now described in the figure legend of Extended Data Fig. 1 that little diffusion is observed upon incubation of cohesin with DNA at a salt concentration of 50 mM, suggesting that the diffusion coefficient of cohesin on DNA is indeed dependent on salt concentration. We attempted to gather a set of data that will allow us to determine the diffusion coefficient of cohesin at 150 mM salt; unfortunately this is time consuming and we are unable to include the data at this point.

4. Referee #2 noted that the references to the figure panels in which oxygen scavenger systems were used are incorrect.

We thank the referee for pointing out this inaccuracy which we have corrected.

5. The referee asks how many molecules were analysed in the histogram in Figure 2b.

We have added this information to the figure legend of Figure 2b. Data was collected from 290 regions that were identified as having a fluorescent signal and 228 background regions that did not contain fluorescence.

6. The referee asks that we show the MSD of a number of different cohesin molecules and asks why the graph originally shown in Figure 2d reaches a plateau at $\sim 14 \mu\text{m}^2$.

We agree with the referee that this is an important point and one that was not adequately described in the manuscript. We have replaced Fig 2d with a new panel in which the diffusion coefficients of cohesin \pm ATP and an ATP binding and hydrolysis deficient form of cohesin are compared. Figure 2d represents the combined data on the diffusion of individual cohesin molecules (cohesin wt – ATP: 13 molecules, cohesin wt + ATP: 18 molecules, cohesin KA mutant + ATP: 28 molecules). We found that MSD plots of individual cohesin complexes show strong fluctuations and cannot be well approximated by normal or bound diffusion. This indicates that at the level of individual complexes statistical fluctuations dominate the random movement and do not allow the reliable extraction of diffusion coefficients of single diffusing molecules. Therefore, we combined data from individual molecules and calculated the average MSD value for each time-point (Extended Data Fig. 2f). Unlike the individual plots, the combined plot shows behaviour typical for bound diffusion and allows for the extraction of the diffusion coefficient during normal diffusion by determining the slope of the linear part of the graph (data plotted in Fig. 2d). Therefore, in the revised manuscript we decided to use only data combined from single cohesin molecules in order to calculate diffusion coefficients.

As also pointed out by the referee, we indeed observe that the MSD plot plateaus. Bound diffusion is typical in systems where diffusion is confined to a certain area. In our case, cohesin molecules undergo one-dimensional diffusion along doubly tethered DNA molecules and the ends of the DNA immobilized on the coverslip naturally restrict their movements. We also observed that other cohesin complexes present on DNA, which decreased effective area allowed for the diffusion further, could also restrict the diffusion of cohesin molecules. We have added discussion of this point to the figure legend of Extended Data Fig. 2f.

7. Referee #2 comments that our estimate as to the diameter of the QDot antibody conjugates used in Figure 3 is inaccurate, and questions our assertion that nucleosomes are unlikely to act as a barrier to cohesin movement.

We thank the referee for highlighting this inaccuracy; we have modified the text accordingly. In addition, we have performed new roadblock experiments using recombinant nucleosomes. These data show that, unlike QDotEcoRIE111Q, a nucleosome does not act as a non-diffusible barrier to cohesin (Fig. 3c).

8. The referee states that the evidence provided in Extended Data Fig. 1f is insufficient to claim that cohesin molecules do not pass one another, and suggests an experiment in which two differently coloured forms of cohesin are used to test this idea more rigorously.

We acknowledge that the data provided is insufficient to claim that two cohesin molecules never pass one another on DNA. Unfortunately due to the technically challenging nature of dual colour single molecule imaging we were unable to generate sufficient data to include it in this version of the manuscript and so have removed this statement from the text

9. Referee #2 suggests that we use RNase to test whether cohesin translocation is affected by T7 RNAP directly or the presence of an RNA transcript.

We thank the referee for suggesting this very interesting experiment. We agree that it is likely that the RNA transcript rather than T7 RNAP *per se* acts as a roadblock to cohesin translocation and have added a sentence to the text to highlight this possibility. Unfortunately, we were unable to perform this experiment due to time constraints.

Referee #3

1. Referee #3 comments that we do not accurately report the findings of Murayama and Uhlmann (Murayama & Uhlmann, Nature 2014) and asks us to determine the ATP requirements of cohesin loading in our bulk and TIRF assays.

We thank Referee #3 for highlighting this issue. We have modified the text to more clearly state the findings of Murayama and Uhlmann (namely that DNA cleavagesensitive DNA binding of fission yeast cohesin was not tested in the absence of ATP but was demonstrated in the absence of Scc2/4). Furthermore, we have performed a set of experiments to determine the role of ATP on the salt resistant binding of cohesin to DNA in our bulk and TIRF microscopy assays. We find that the presence of ATP is not required for salt-resistant cohesin:DNA interactions (Extended Data Fig. 2c, d) and does not measurably affect the diffusion coefficient of single cohesin complexes (Fig 2d, Extended Data Fig. 2e, f). We have also tested the DNA binding of an ATP binding/hydrolysis deficient form of cohesin (Smc1/3 KA). KA cohesin reproducibly bound less DNA than wildtype cohesin in our bulk assay, suggesting that ATP binding/hydrolysis might enhance but is not required for cohesin:DNA interactions in vitro. KA cohesin also bound to DNA in our TIRF microscopy assay and the diffusion coefficient of single molecules of KA cohesin was indistinguishable from wildtype cohesin.

2. Referee #3 comments that the reference to photobleaching data in Extended Data Figure 1f is confusing.

We have modified the text accordingly.

3. The referee suggests that the mitotic transcription-dependent restriction of Sgo1 and possibly cohesin at the inner centromere can be explained by transcription-dependent sliding, and suggests that we discuss this possibility in the text.

We thank Referee #3 for this interesting suggestion. We have added a sentence to the discussion section and cited the relevant paper.

Thank you for submitting your manuscript for consideration by the EMBO Journal. We have evaluated the manuscript, the three previous referee reports and your responses from both rounds of review.

We are happy in principle to publish the paper with minimal delay, but need to ask you to address a few issues before formal acceptance:

- 1) Stigler et al, Cell Reports has to be formally cited and described/compared in your paper.
- 2) Please update us if the following experiments, which were explicitly requested by the referees, have by now been done - if so, we would very much like to include them in the revised study:
 - a) T7RNA pol. inhibitor or catalytically inactive T7 pol mutants (ref No. 1). Similar, RNase treatment (ref no. 2, point 9).
 - b) dependence of diffusion coefficient on salt concentration (ref no. 2: you mentioned in the rebuttal that you have data at 150 mM salt: can that be included please?
 - c) fig 1 in rebuttal to ref no. 2 (tether length) - if not already included in the revision, please do so.
 - d) ref no. 2, point 6: MSD for individual cohesin molecules. We appreciate you note the statistical fluctuation. As per the journal's source data policy, please include the individual data as 'source data' files, which will be provided as linkouts from the figure.

As per journal policy, please include source data files for all key experiments, which will be provided as linkouts from the figures. Please see the journal guide to authors for further information (see also 2d above).

We also require a synopsis test (a short standfirst followed by 3-5 bullet points to highlight key findings in the paper; the text should be complementary to the title+abstract). Please also ensure that you have included up to three keywords that describe the dataset best.

Please note that as part of the EMBO transparent process we will include your response to the referee comments [since the manuscript was transferred, not the primary referee comments]. This has been cleared with Nature. Please therefore include the actual referee points addressed in the rebuttal in *Italics* followed by your current responses from both rounds of review. In this way the text makes sense to the interested reader. The alternative is to opt out of this policy, but only <2% authors do so.

For more details on our Transparent Editorial Process, please visit our website:
http://emboj.embopress.org/about#Transparent_Process

Thank you for the opportunity to consider your work for publication in the next week. I look forward to your revision and rapid publication.

Corresponding Author Name: Jan-Michael Peters
Journal Submitted to: EMBO Journal
Manuscript Number: